# Reconstructing dynamics of the Baltic Ice Stream Complex during deglaciation of the Last Scandinavian Ice Sheet

Izabela Szuman[1], Jakub Z. Kalita[1], Christiaan R. Diemont[2], Stephen J. Livingstone[2], Chris D. Clark[2], Martin Margold[3]

[1]Department of Geomorphology, Adam Mickiewicz University, Poznań, 62-712, Poland
[2] Department of Geography, Sheffield University, Sheffield, S10 2NT, United Kingdom of Great Britain - England, Scotland, Wales
[3] Department of Physical Geography and Geoecology, Charles University, Prague, 128 43, Czech Republic

*Correspondence to*: Izabela Szuman (szuman@amu.edu.pl)

**Abstract.** Landforms left behind by the last Scandinavian Ice Sheet (SIS) offer an opportunity to investigate controls governing ice sheet dynamics. Terrestrial sectors of the ice sheet have received considerable attention from landform and stratigraphic investigations. In contrast, despite its geographical importance, the Baltic Sea remains poorly constrained due to limitations in bathymetric data. Both ice sheet scale investigations and regional studies at the southern periphery of the SIS have considered the Baltic depression as a preferential route for ice flux towards the southern ice margin throughout the last glaciation. During the deglaciation the Baltic depression hosted the extensive Baltic Ice Lake, which likely exerted a considerable control on ice dynamics. Here we investigate the Baltic depression using newly available bathymetric data and peripheral topographic data. These data reveal an extensive landform suite stretching from Denmark in the west to Estonia in the east and from the southern European coast to the Åland Sea, comprising an area of 0.3 million km$^2$. We use these landforms to reconstruct aspects of the ice dynamic history of the Baltic sector of the ice sheet. Landform evidence indicates a complex retreat pattern that changes from lobate ice margins with splaying lineations, to parallel Mega-Scale Glacial Lineations (MSGLs) in the deeper depressions of the Baltic Basin. Ice margin still-stands on underlying geological structures indicate the likely importance of pinning points during deglaciation resulting in a stepped retreat signal. Over the span of the study area we identify broad changes in ice flow direction, ranging from SE-NW to N-S and then to NW-SE. MSGLs reveal distinct corridors of fast ice flow (ice streams) with widths of 30 km and up to 95 km in places, rather than the often-interpreted Baltic-wide (300 km) accelerated ice flow zone. These smaller ice streams are interpreted to have operated close behind the ice margin during late stages of deglaciation. Where previous ice sheet-scale investigations inferred a single ice source, our mapping identifies flow and ice marginal geometries from both Swedish and north Bothnian sources. We anticipate that our landform mapping and interpretations may be used as a framework for more detailed empirical studies by identifying targets to acquire high resolution bathymetry and sediment cores and also for comparison with numerical ice sheet modelling.

## 1 Introduction

The Baltic depression (Fig. 1), currently occupied by the Baltic Sea, was a defining feature of the bed of the Scandinavian Ice Sheet (SIS) likely exerting considerable control on the direction of ice flow and position of ice margins during the last glaciation (Holmlund and Fastook, 1995; Patton et al., 2017; Patton et al., 2016; Stroeven et al., 2016). The depression has an amplitude of 200 to 300 m (Fig. 1D), and is positioned such that it might have steered ice evacuation from the main ice divide towards ice margins in the south. The topographic steering of ice flow may have been further enhanced by unconsolidated sediments, previously accumulated in the depression, reducing the basal shear stress and promoting faster ice flow velocities compared to surrounding bedrock shield areas (Amantov et al., 2011; Boulton and Jones, 1979; Boulton et al., 1985). Glacio-isostatic effects of ice loading increased the reverse-slope setting of the southern Baltic, impeding water evacuation from the proglacial zone (Larson et al., 2006; Uścinowicz, 2003). The ice margin retreating across the Baltic depression from its Last Glacial Maximum (LGM) position was thus water-terminating, impacting ice dynamics due to calving and sub-aqueous melting (e.g., Noormets and Flodén, 2002b, a; Uścinowicz, 1999). Ice dynamics and changes in ice flow direction were also linked to ice divide shifting from the Scandinavian mountains to over the Bothnian Basin (Boulton et al., 1985; Kleman et al., 1997; Patton et al., 2016).

Despite the factors, outlined above, that might have influenced ice geometry and dynamics, much of the glacial geomorphological record within the Baltic depression remains undocumented because it is obscured by the modern-day Baltic Sea. There was no submarine evidence available at the time Boulton et al. (2001) reconstructed the ice stream pattern along the Baltic (Fig. 2); the aim of this paper is to seek information about the footprint of the ice stream and its likely width. We use a suite of newly available bathymetric and coastal datasets to map the glacial geomorphology across a ~310,000 km$^2$ region, encompassing the current Baltic Sea floor and adjoining coastal areas (Fig. 1B). We present a consistent sector-wide glacial landform map of the Baltic depression and use these data to reconstruct past ice flow geometries and major ice margin positions. Specifically, we use the mapped landform record to address the following questions:

1. As has frequently been suggested from empirical and modelling investigations (Boulton et al., 1985; Patton et al., 2017; Patton et al., 2016; Punkari, 1997; Stephan, 2001), does the landform record support a depression-wide (300 km) Baltic Ice Stream during the LGM?

2. What was the character of ice margin retreat though the Baltic depression? Is there evidence of water-terminating calving ice fronts or lobate terrestrial-style margins?

3. To what extent did topographic pinning points arising from geological structures or lithological contrasts influence ice marginal retreat?

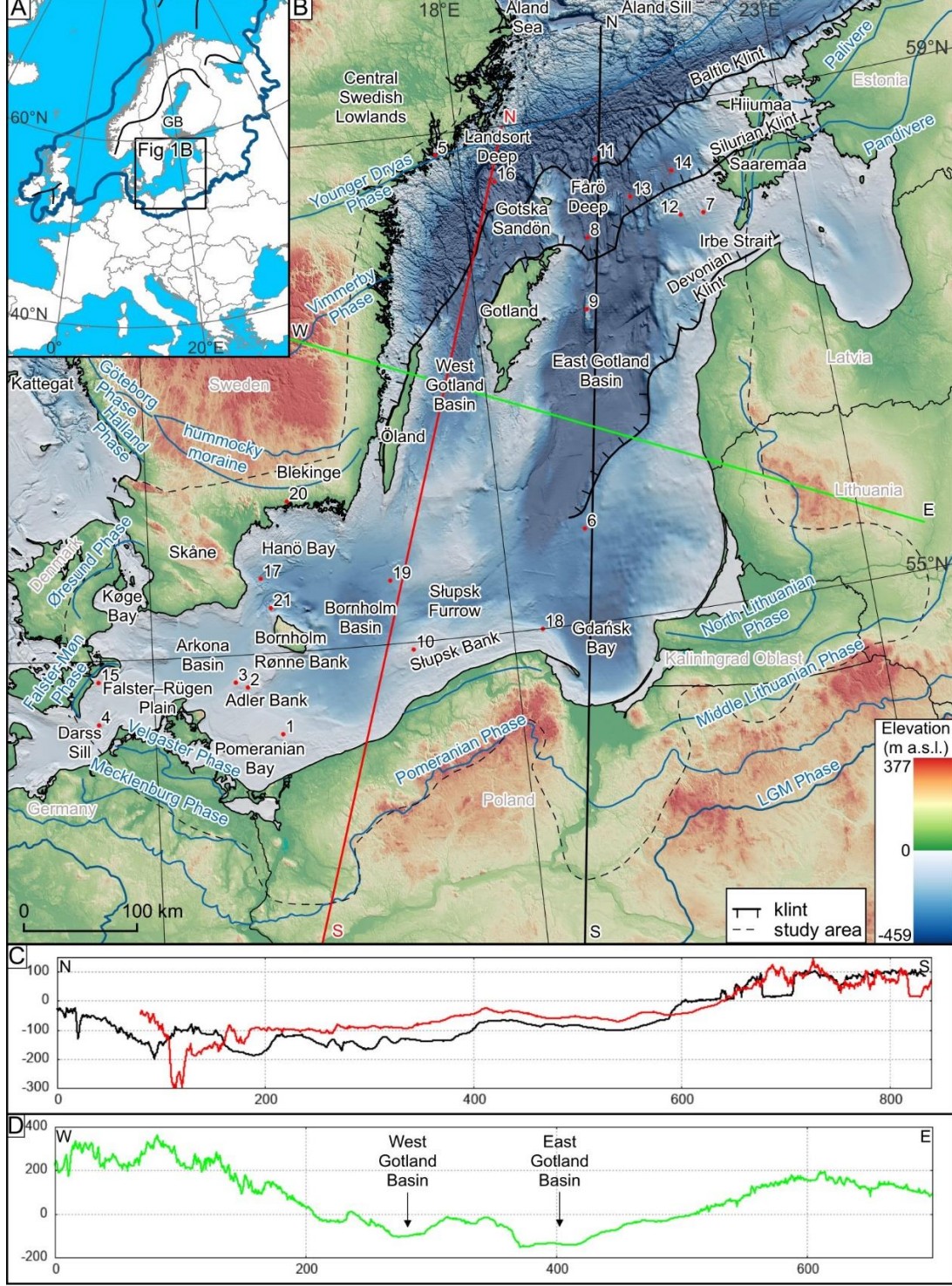

**Figure 1: (A) Last Glacial Maximum extent over Europe (thick blue line; Svendsen et al. 2004). GB indicates Gulf of Bothnia, located north of study area. Black line indicates ice divides around LGM (after Kleman et al., 2008; Patton et al., 2016). The base map is**

from EuroGeographics and UN-FAO. (B) Study area, which covers the Baltic Sea and nearby coastal zone. Major phases are drawn in blue, with their names indicated. The Baltic Klint constitutes a margin between crystalline bedrock (to the north), and Ordovician sedimentary rocks (to the south). The Silurian Klint separates Ordovician (north) and Silurian rocks (south). Red dots and associated numbers indicate key previous studies of offshore glacial landforms (see Table 1). The DEM is based on EU-DEM and EMODnet (see Table 2). (C) The N-S profiles (see the black and red line in panel B) reveals a general upslope toward the south, which would have been even steeper during glaciation due to glacio-isostatic loading, and cuesta-like topography associated with geological steps at the Baltic Sea bottom. (D) The NW – SE profile (see the green line in panel B) depicts the ~ 400 km wide on-and-offshore depression divided by Gotland Island into two troughs: eastern and western along the East and West Gotland basins respectively.

**Table 1: Key previous studies on offshore glacial landforms within the study area**

| Number in Fig. 1B | Localisation | Type of investigations | Referencess |
|---|---|---|---|
| 1 | Odra Bank | Stratigraphy based on sediment cores and radiocarbon dating | (Kramarska, 1998) |
| 2 | Adler Grund | Interpretation of glacial landforms based on hydroacoustic survey and seismic profiles | (Feldens et al., 2013) |
| 3 | SE Arkona Basin | Stratigraphy of glacial deposits inferred from seismic profiles and sediment cores | (Obst et al., 2017) |
| 4 | Darss Sill | Stratigraphy of sediments based on hydroacoustic survey and sediment cores | (Lemke and Kuijpers, 1995) |
| 5 | Stockholm Archipelago | Interpretation of landforms based on hydroacoustic and sub-bottom profiles | (Jakobsson et al., 2016) |
| 6 | Gdańsk-Gotland Sill | Analysis of iceberg and ice-keel ploughmarks based on hydroacoustic survey and sediment sampling | (Dorokhov et al., 2018) |
| 7 | W of Estonia | Interpretation of glacial deposits and landforms based on seismic profiles | (Noormets and Flodén, 2002a) |
| 8 | Around Fårö Deep | Interpretation of glacial deposits and landforms based on seismic profiles | (Noormets and Flodén, 2002b) |
| 9 | Klints Bank | Interpretation of seismic profiles | (Schäfer et al., 2021) |
| 10 | Southern Baltic | Interpretation of glacial landforms based on seismic profiles | (Uścinowicz, 1999) |
| 11 | The Baltic Klint | Interpretation of seismic profiles | (Tuuling and Flodén, 2016) |
| 12 | Silurian reefs, between Saaremaa and Gotland | Interpretation of seismic profiles | (Tuuling and Flodén, 2013) |
| 13 | between Gotland and Saarema | Interpretation of seismic profiles | (Tuuling and Flodén, 2001) |
| 14 | W of Estonia | Analysis of iceberg scours based on hydroacoustic survey | (Karpin et al., 2021) |
| 15 | Falster-Møn area | Interpretation of glacial deposits based on hydroacoustic survey and seismic profiles | (Jensen, 1993) |
| 16 | North of Gotland | Interpretation of seismic profiles | (All et al., 2006) |
| 17 | Hanö Bay | Stratigraphy of glacial deposits inferred from seismic profiles and sediment cores | (Björck et al., 1990) |
| 18 | Southern Baltic | Interpretation of glacial landforms based on seismic profiles | (Uścinowicz, 2003) |
| 19 | Arkona Basin, East Baltic | Analysis of glacial valleys (seismic profiles) | (Flodén et al., 1997) |
| 20 | Hanö Bay Stockholm Germany | Interpretation of hydroacustic survey | (Jakobsson et al., 2020) |
| 21 | offshore Bornholm | Stratigraphy based on seismic profiles | (Perini et al., 1996) |

## 2 Background: the role of the Baltic depression during the Last Glacial

The effects of the Baltic depression on the dynamics of the Last SIS have been discussed since the late 19[th] century (Madsen, 1898; Zeise, 1889; after Stephan, 2001). This includes evidence for a major, up to 130 degrees, switch in ice flow direction, from towards the south to west and north-westward in the region of the southern Baltic and Denmark. The north-westward and westward flow into Denmark, after the LGM, has subsequently been termed the Young Baltic Advances (Holmström, 1904; in Glückert, 1974; Stephan, 2001) (Fig. 2, arrow B1). Subsequently, it was hypothesised that this change in ice flow direction in the southern sector of the ice sheet was due to an eastward shift of the main ice divide (Ahlmann et al., 1942; Eissmann, 1967; Enquist, 1918; Ljunger, 1943; Woldstedt and Duphorn, 1974). An alternative explanation for the flow directional changes was that the upland topography behind the German and Polish coasts caused ice to be redirected westward (e.g., Gripp, 1981 after Stephan, 2001). Such behaviour could be a consequence of increasing topographic dependence during deglaciation (Kjær et al., 2003). Lithostratigraphy and striae analysis by Ringberg (1988) indicated that reduced ice supply over Sweden caused the Baltic Ice Stream to turn westwards into Denmark when the ice sheet was retreating from its maximum extent (Fig. 2, B1). The westward ice flow redirection was also modelled by Boulton et al. (1985). North-westward flow into Skåne and Denmark has also been suggested for the period prior to the maximum ice sheet extent (termed the Old Baltic Advance, Ringberg, 1988), based on the lithostratigraphy of till deposited in Denmark and from striae orientations.

Boulton et al. (2001) conducted one of the earliest satellite-driven mapping campaigns of glacial lineations at the ice sheet scale, resulting in possible ice stream configurations during deglaciation (Fig. 2). Subsequent landform investigations using higher resolution data identified potential palaeo-ice streams in the marine and terrestrial record within- and surrounding the Baltic Sea, based on the identification of Mega-Scale Glacial Lineations (MSGLs) (Greenwood et al., 2016; Jögensen and Piotrowski, 2003; Kalm, 2012; Szuman et al., 2021; Greenwood et al. 2023). These ice stream imprints commonly conform to local topographic depressions, indicating that topography played a role on flow geometry, at least regionally, during ice retreat.

As our observational record of modern ice sheet dynamics expanded, so did the idea that the Baltic depression facilitated an ice stream that operated throughout the last glacial (Holmlund and Fastook, 1995; Patton et al., 2017; Patton et al., 2016; Stroeven et al., 2016). Numerical ice sheet modelling efforts focused on the SIS typically input basal shear stress values based on weak substrates and an assumed existence of ice streams here (Holmlund and Fastook, 1995; Patton et al., 2017; Patton et al., 2016), resulting in synchronous, Baltic-wide ice streaming in the low-shear-stress regions. It therefore remains difficult to determine whether the Baltic ice streams suggested by some of these earlier studies are realistic.

Sector-scale landform mapping has been carried out for the Bothnian Basin from high resolution bathymetry data (Greenwood et al., 2015; Greenwood et al., 2016). These data revealed early Holocene ice streaming, documented by cross-cutting glacial lineations and an abundance of meltwater landforms and crevasses-squeeze ridges (e.g., Greenwood et al., 2016). In contrast, patchy bathymetric data has until very recently (see Greenwood et al., 2023 for an independent study of the same area) limited

investigations in the Baltic depression to localized reconstructions and extrapolation from nearby terrestrial records (Boulton et al., 2001; Cuzzone et al., 2016; Kleman et al., 1997), mixed with inferences from glaciological intuitions and hunches (Fig. 1).

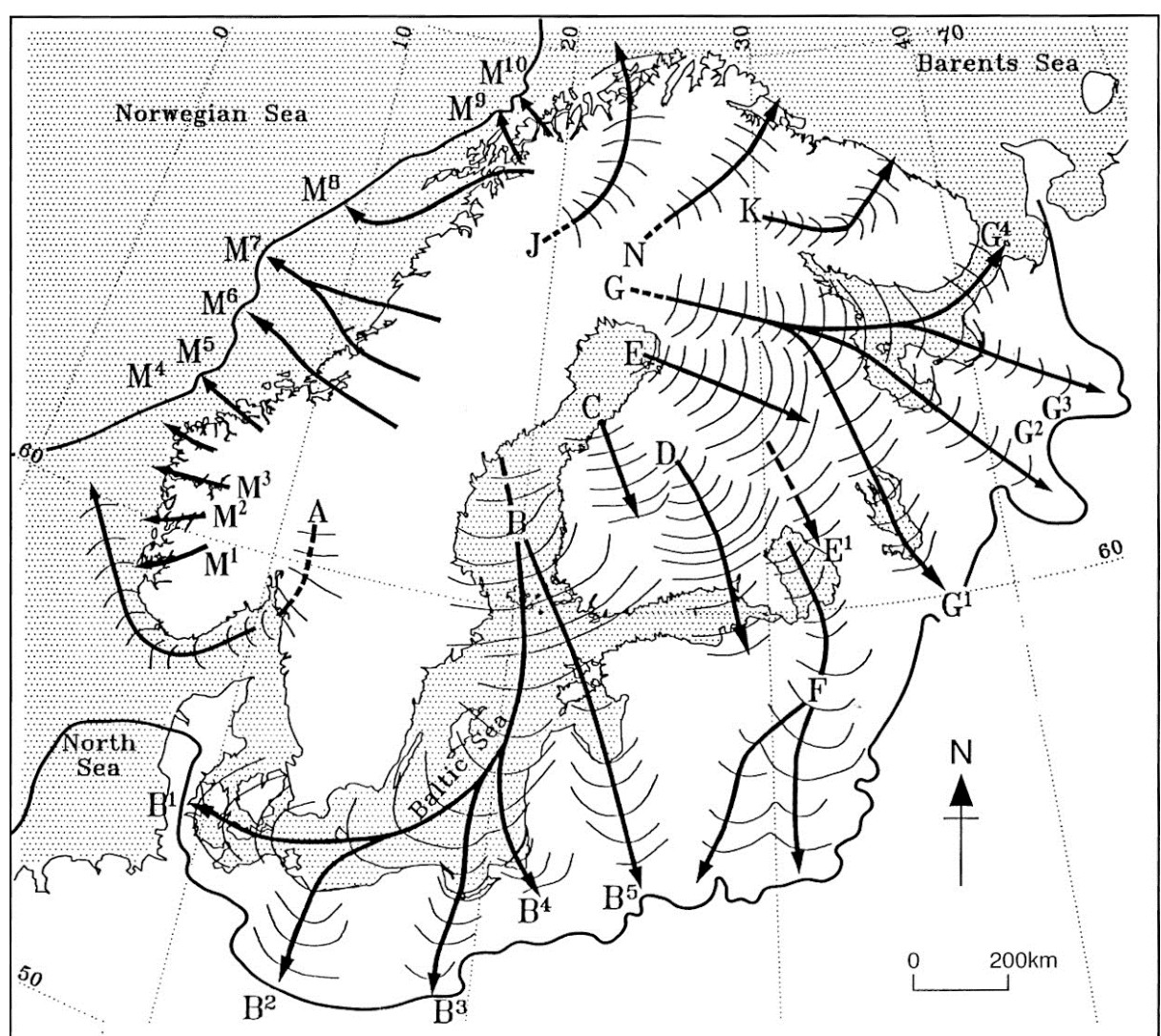

**Figure 2. An early depiction of time-transgressive ice stream trajectories with associated back-stepping ice margins after Boulton et al. (2001; Reprinted from Quaternary Science Reviews, 20/9, Boulton, G.S., Dongelmans P., Punkari, M., Broadgate, M., Palaeoglaciology of an ice sheet through a glacial cycle: the European ice sheet through the Weichselian, 591-625, Copyright (2001), with permission from Elsevier). The Baltic Sea depression is inferred to control the ice stream location but note that this depiction does not imply a synchronous Baltic Ice Stream of this length (>1000 km) branching into B1 to B4. Instead, it depicts numerous short length (<300 km) ice streams on land (B1 to B4) that, once back-stepped into the Baltic depression, eventually merged into a single ice stream. No submarine evidence was available at the time Boulton et al. (2001) drew the figure and it was the purpose of this paper to seek information about the footprint of this ice stream and its likely width. The relative timing of the ice streams draining the Baltic depression**

**has been established only partially, with Kjær et al. (2003) and Houmark-Nielsen and Kjær (2003) noting that B1 (called the Young Baltic Advances) came after the B2 lobe into Germany and Poland.**

The bathymetry of the Baltic Sea, with its prominent N-S and E-W trending depressions, has been attributed to the geological structures of the region and its differential erosion by Pleistocene ice sheets (Hall and Van Boeckel, 2020; Poprawa et al., 1999). The north-western Baltic Basin has a crystalline Proterozoic substrate typical of the Scandinavian Shield and where sediment cover is limited, bedrock structure is clearly visible on bathymetric data (Fig. 1B). Stratigraphic and seismic surveys

have indicated that the sets of prominent NE-SW oriented ridges ('Klints') between Estonia and Sweden correspond to lithological transitions between Cambrian, Ordovician, and Silurian rocks, controlling erodibility and generating stepped morphologies (Fig. 1C) (Tuuling and Flodén, 2016). The SE Baltic Basin is dominated by more readily erodible lithologies with Paleogenic and Neogenic sandstones and mudstone sequences. Sediments thickness and composition varies through the Baltic with the northern part having patchier cover, between 20 and 30 m in thickness, while sediment cover further south is

more extensive with thicknesses of up to 60 m (Björck et al., 1990; Noormets and Flodén, 2002b, a; Uścinowicz, 1999). The surficial deposits comprise glaciogenic sands and lacustrine sediments as well as marine strata (Sviridov and Emelyanov, 2000). Zones of unconsolidated sediment accumulation occur throughout the whole Baltic Sea. Those, in the northern sector are composed of glaciofluvial material and were interpreted as subglacial outwash deposits and ice marginal grounding line deposits (Noormets and Flodén, 2002b). Accumulations of unconsolidated glacifluvial and glacial sediments particularly

correlate with the position of sills and banks (Kramarska, 1998; Lemke and Kuijpers, 1995; Obst et al., 2017; Uścinowicz, 1999; Noormets and Flodén, 2002a).

**3 Methods**

In this study, we map landforms in submarine and peripheral coastal regions (Fig. 1B) of the present-day Baltic Sea. The

145 offshore area coincides with so-called Baltic Proper, an area extending from the Åland Sill in the north to Darss Sill in the south-west, and Hiiumaa and Saaremaa islands in the east. Our motivation for mapping the coastal periphery of the Baltic is twofold:

     1. It eases integration with the much-studied onshore landform record,

2. To provide a high-resolution verification for ice-flow patterns and ice-marginal retreat patterns mapped from the lower quality and resolution bathymetric dataset.

Bathymetric data varied substantially in resolution and quality across offshore regions. We used the Digital Bathymetry Model of the European Marine Observation and Data Network v. 2018 and 2022 (EMODnet Bathymetry Consortium, 2022). Its

source data is a conglomerate of data sets with resolutions typically between 20 to 50 m but with some regions of 200 to 500 m resolution (Fig. 3) (Jakobsson et al., 2019). Confident identification of glacial landforms was not possible in the low-

resolution regions. Artifacts are common in the dataset (see e.g., Fig. 3B) and were cross-checked with hydroacoustic and seismic data (Table 1) were available. High resolution (1-5 m) Digital Elevation Models (DEM) were used to map the peripheral terrestrial regions (Table 2), with the exception of the Kaliningrad region, where a 25 m resolution DEM was used.

Mapping was carried out manually using the open-source Quantum Geographic Information System (QGIS). Semi-transparent hillshades, with numerous azimuth angles to limit visualization bias (Smith and Clark, 2005) were superimposed on the DEM. Vertical exaggerations between 1 and 40 assisted in identification of low amplitude landforms, with cross-profiles and colour palette manipulation proving a powerful tool for verifying landforms, especially in regions of low resolution and quality of data. When possible, geological maps (Table 3) were used to support our interpretations, although the resolution typically

exceeds the size of the landforms, and only provides an indication of surficial sediments. Our approach was focused on the geomorphological expression of landforms not on their surface geology e.g., if a ridge met the characteristics of a moraine ridge (fan shaped, perpendicular to the ice flow direction and, in ideal situation, associated with other landforms) it was judged as such even if it was covered by for example, marine sands.

**Table 2. The elevation data sources used in this study.**

| Region | DEM spatial resolution [m] | Provider | Service |
|---|---|---|---|
| **Baltic Sea** | Dataset: 50 - 115 <br> Source: 20 - 500 | EMODnet, 01.2019, 12.2023 | emodnet.ec.europa.eu |
| **Kaliningrad** | 25 m | EU-DEM, 01.2019 | eea.europa.eu |
| **Sweden** | 1 | Lantmäteriet, 11.2020 | lantmateriet.se |
| **Denmark** | 1 | Styrelsen for dataforsyning og effektivisering, 11.2020 | dataforsyningen.dk |
| **Germany, Brandenburg** | 1 | © GeoBasis-DE/LGB, dl-de/by-2-0, 11.2020 | geobasis-bb.de |
| **Germany, Mecklenburg-Vorpommern** | 1 | © GeoBasis-DE/M-V 2020 | laiv-mv.de |
| **Poland** | 1 | GUGiK, 07.2020 | geoportal.gov.pl |
| **Lithuania** | 5 | © Nacionalinė žemės tarnyba prie Žemės ūkio ministerijos, GDR50LT, 22.07.2021 | geoportal.lt |
| **Latvia** | 1 | Latvijas Ģeotelpiskās informācijas aģentūra, 11.2020 | latvija.lv |
| **Estonia** | 1 | Estonian Land Board 2021 | maaamet.ee |

**Table 3. Geological data sources used in this study (all accessed 12.2023).**

| Region | Institution | Service |
|---|---|---|
| **Baltic Sea** | EMODnet | emodnet.ec.europa.eu |
| **Sweden** | Geological Survey of Sweden | www.sgu.se |
| **Denmark** | Geological Survey of Denmark and Greenland | geus.dk |
| **Germany** | Federal Institute for Geosciences and Natural Resources | geoportal.bgr.de |
| **Poland** | Polish Geological Institute | pgi.gov.pl |
| **Lithuania** | Lithuanian Geological Survey | inspire-geoportal.lt |
| **Latvia** | LVM Geo | lvmgeo.lv |
| **Estonia** | Republic of Estonia Land Board | maaamet.ee |

Glacial lineations were mapped along the crest-line using polylines and included features in bedrock and unconsolidated sediments such as MSGLs, drumlins, crag-and-tails and grooves. Given their linear nature and similarity to some known artefacts in low resolution data, lineations were subdivided into different confidence levels ('glacial lineations' and 'uncertain glacial lineations'). Ribbed moraines and regions of iceberg ploughmarks were mapped as polygons. We also mapped ice-contact landforms as polylines along their highest elevated area and with two confidence levels. Ice-contact landforms encompass moraines, grounding zone wedges (GZW), and ice-contact fans. All are oriented perpendicular to the ice flow direction. Moraines have the most prominent fan-shaped ridge with a roughly symmetrical cross-profile. In contrast, GZWs have a steep distal and gentle proximal slope with a sinusoidal pattern of ridges, that often occur in swarms. Ice contact-fans have steep proximal and gentle distal slope. Eskers were mapped along the crest-line and grouped into two confidence levels. Meltwater channels were mapped by polylines along their lateral margins and classified into tunnel valleys, marginal meltwater channels and uncertain meltwater channels. Tunnel valleys were distinguished based on their orientation parallel to glacier flow. They are often associated with eskers, terminate near former ice margins, and have undulating long profiles (Kehew et al., 2012). In some offshore localities associated with the low-resolution DEM or unclear situation (e.g., lack of eskers) tunnel valleys were mapped as uncertain. Marginal meltwater channels often form a swarm of parallel channels that document the position of former glacier margins (Margold et al., 2013). We did not map meltwater channels along the Swedish coast, from Blekinge towards Central Swedish Lowlands and northern Baltic where the hard bedrock geology contains numerous fissure valleys with complicated patterns, with only some occupied by eskers.

The mapped glacial lineations formed the basis for developing flowsets (Boulton and Clark, 1990; Clark, 1990; Greenwood and Clark, 2009). A flowset comprises a coherent set of morphologically and directionally associated lineations, interpreted to represent a discrete ice flow event. Low-resolution data in regions such as the eastern part of East Gotland Basin and large parts of the southern Baltic (i.e., Arkona and Bornholm basins, Słupsk Furrow and Gdańsk Bay; Figs 1B, 3) resulted in gaps

between regions of higher density mapping. However, it was often possible to detect larger features in these regions by investigating multiple variants of hill-shade angles, z-factors, colour palettes, terrain profiles and comparing them with available studies (Table 1).

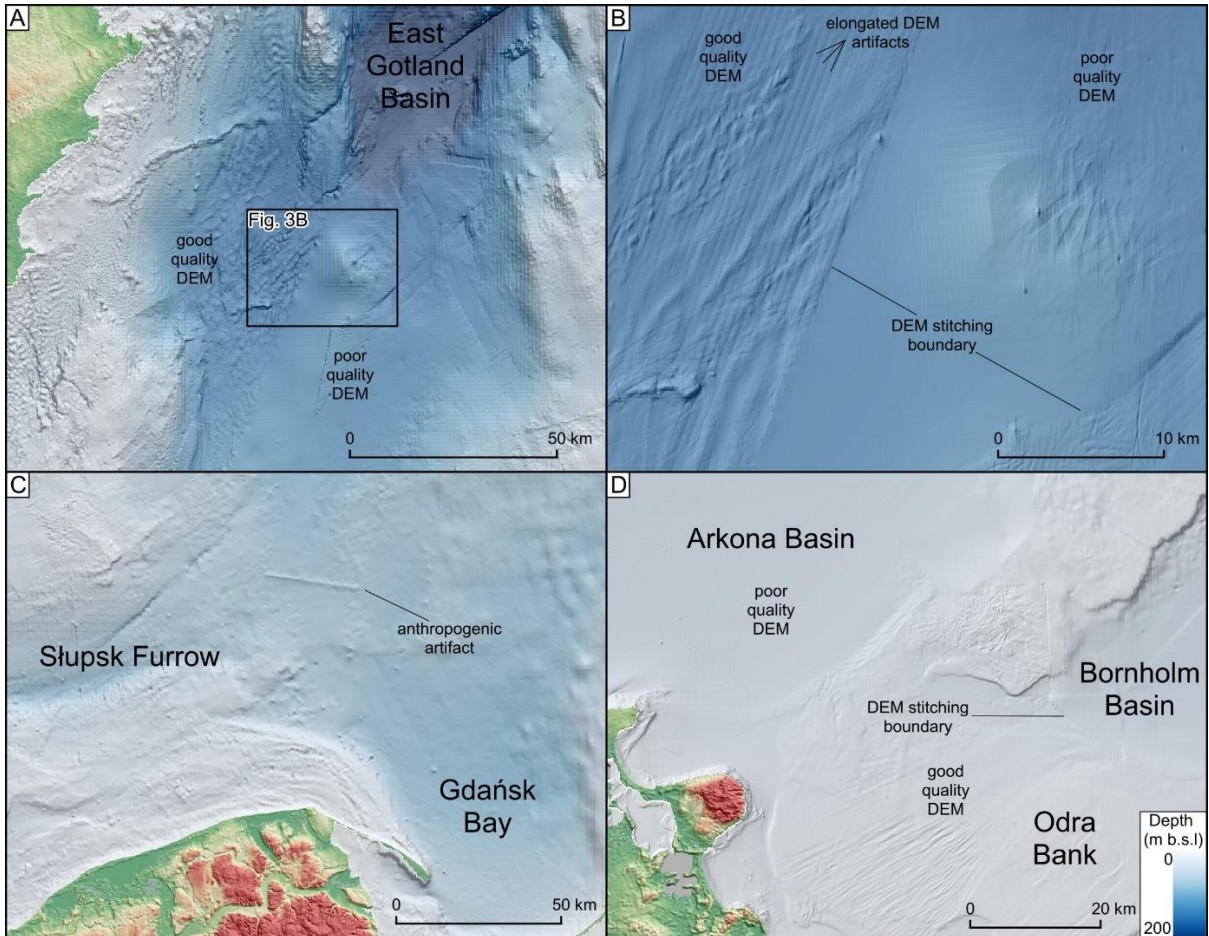

**Figure 3: Examples of DEM artifacts and high- and low-resolution bathymetric data. The DEM is based on EMODnet (see Table 2). (A) Higher resolution bathymetric data for the western part of the East Gotland Basin enabled landform recognition, whereas the low-resolution data for the eastern part made it difficult to identify glacial landforms. The onshore DEM is based on the EU-DEM (see Table 2). (B) Examples of DEM artifacts in low- and high-resolution data, especially visible along the DEM stitching boundary. Note the difficulty in identifying landforms in the low-quality data region. (C) Low resolution DEM for southern Baltic, with landforms not distinguishable in Gdańsk Bay but slightly more visible along the Słupsk Furrow. The onshore DEM was provided by GUGIK (see Table 2). (D) The contrasting data quality for the high-resolution DEM of Odra Bank vs the low-resolution DEM along the Arkona and Bornholm basins. The onshore DEM was provided by © GeoBasis-DE/M-V 2020 (see Table 2).**

## 4 Results - Geomorphology and landforms of the Baltic Sea area

About 25,000 features were mapped within the study area (Fig. 4). Most of the regions with higher resolution data (both marine and terrestrial) display glacial landforms. The offshore features constitute 25% of the total object population. The highest density of landforms is found along the Swedish, German, Polish, and Lithuanian coast, and in the vicinity of the area between

Öland and Gotland islands. The lowest density of landforms is along the central and southern Baltic Sea, however, the few landforms found here are of considerable size. Glacial lineations comprise ca 15,000 of the mapped features with eskers numbering ca 5,200. The majority (in terms of percentage) of lineations and eskers were mapped in coastal regions. Burial of landforms in marine environments has been identified before (Flodén et al., 1997; Kirkham et al., 2022) and so our mapping is likely an incomplete morphological record in parts of the Baltic depression.

## 4.1 Streamlined bedforms

A variety of glacial streamlined bedforms including MSGLs, drumlins, crag-and-tails and grooves are mapped in the study region (Fig. 5) consistent with mapping from other studies in the region (Noormets and Flodén 2002a; Schäfer et al., 2021). Streamlined bedforms are widely distributed and found in regions of soft sedimentary rocks or areas of thin drift over predominantly bedrock exposed regions. Glacial lineations occur throughout the Baltic Sea to a depth of at least 260 m b.s.l. Almost 75% of the glacial lineations are shorter than 1200 m. They cluster in the terrestrial parts of the Baltic Shield and in the southern sector of the SIS in Poland (Dowling et al., 2015; Hermanowski et al., 2019). MSGLs are found in the deeper depressions and extend in length up to 80 km. Highly parallel MSGLs start abruptly at the geological transition of the Baltic Klint, between the crystalline shield in the north Baltic, and sedimentary rocks to the south (see also Tuuling and Flodén, 2001, 2016). The densest concentration of glacial lineations were mapped in Landsort Deep (Figs 4, 5G), where they are oriented N-S and NW-SE, with the latter dominant. They occupy pre-Quaternary sandstones, contrary to surrounding hard bedrock to the north, east and west (All et al. 2006).

A swarm of 138 grooves occurs between the Klints. The grooves are inter-mixed with drumlins and exhibit high parallel conformity with them (Fig. 5H). The grooves are straight, with an average length of ca 1300 m. In the northern Baltic, the MSGLs are highly parallel, and their density and length increase towards the central parts of the two troughs located either side of Gotland Island. Sparse eskers and meltwater channels with lengths up to 95 km also occur in the troughs. Glacial lineations, up to 5000 m long, in the terrestrial south form local fields typically restricted to drumlinised belts located close to arcuate moraines and marginal meltwater channels. However, these sets often comprise less than 30 lineations.

A concentration of offshore glacial lineations occur in the southern Baltic in Pomeranian Bay (Fig. 7D). This area displays highly elongated lineations with an average length of 4450 m and widths of 200 to 500 m. The lineations are interpreted to be glacial in origin, but with low confidence given their waviness and overlapping nature. They could also be formed by bottom currents or a mixture of glacial and non-glacial processes (i.e., exposed by erosional activity of bottom currents). Klingberg and Larsson (2017) identified lineations eroded by currents west of Öland Island, however they are clear erosional landforms with an anastomosing pattern of grooves and irregular shape (changeable width and general geometry) of 'positive' sections. MSGLs in Pomeranian Bay (Fig. 7D) have more or less the same width along the whole profile and a regular shape. They have a potentially erosional and depositional origin as the ridges clearly overlap each other from different azimuths, and they represent positive and negative landforms. In the vicinity (point 1 in Fig. 1B) there are ridges composed of glacial till with

250 lineation-like cross profiles (Kramarska, 1998), buried in postglacial sands. Further detailed studies are required to confidently interpret the landforms in Pomeranian Bay.

Cross-cutting of streamlined bedforms is common in the study area, especially in Skåne, Germany, Poland, Öland and Gotland, and offshore in the West and East Gotland basins and Pomeranian Bay. Typically, more delicate and shorter glacial lineations overprint more prominent ones. In contrast, the replacement of cross-cutting with overlapping (with the same orientation)

relationships are identified in the eastern part of the study area.

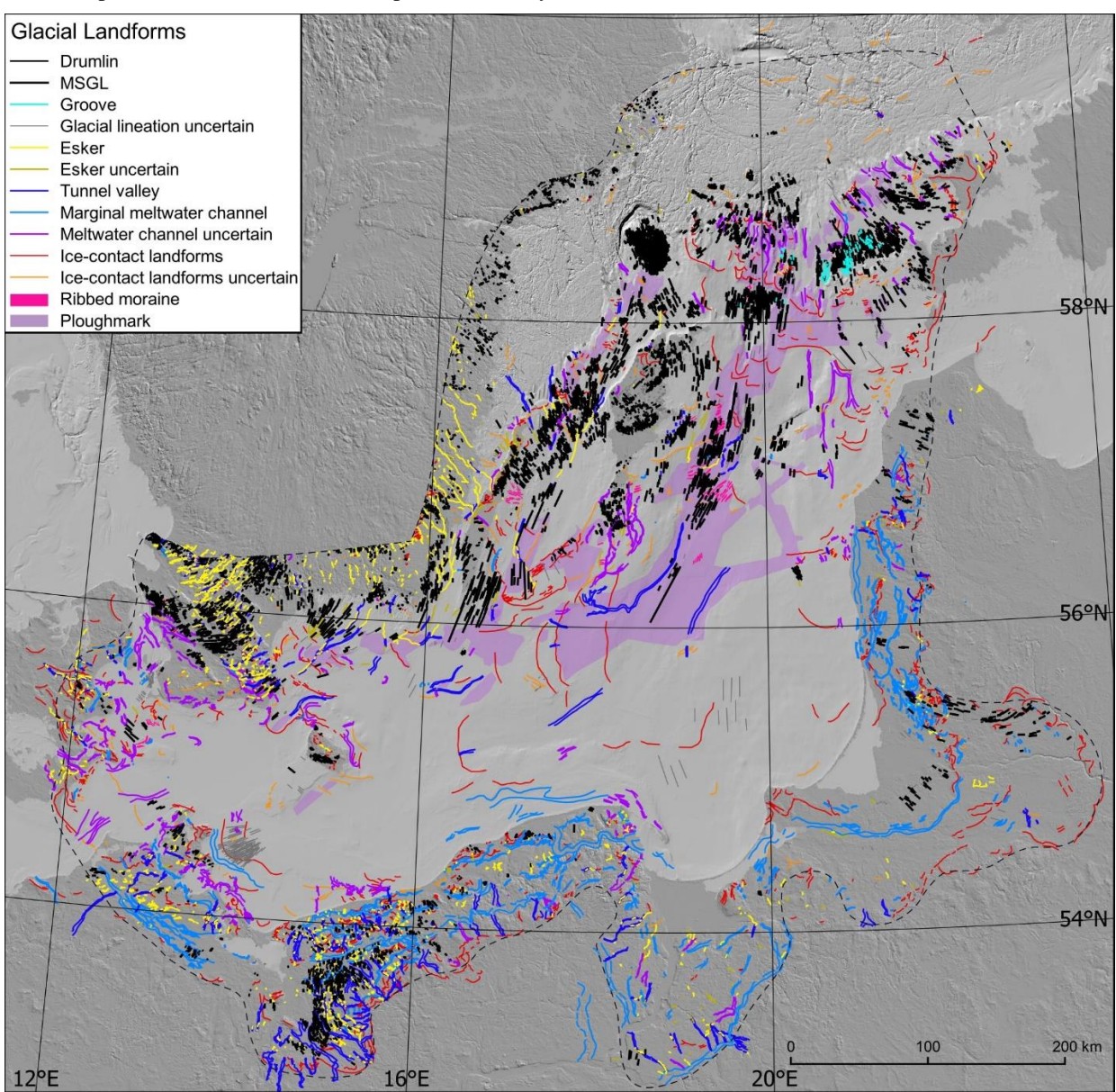

**Figure 4. Landforms mapped in this study. A high-resolution version can be downloaded from the Supplementary Material (Fig. S1). Light and dark grey indicate onshore and offshore areas respectively. Notice the small number of landforms in the offshore northern part of the study area floored by hard bedrock. The DEM is based on EU-DEM and EMODnet (see Table 2)**

## 4.2 Ribbed moraines

Ribbed moraines are identified in the study area in five offshore and one onshore locality (e.g., Figs 5B, I), and typically exhibit a spatial transition downstream into glacial lineations. Ribs are often associated with-, and oriented perpendicularly to topographic highs and obstacles (Fig. 5B). In one locality, in the East Gotland Basin, glacial lineations are superimposed on
ribbed moraines.

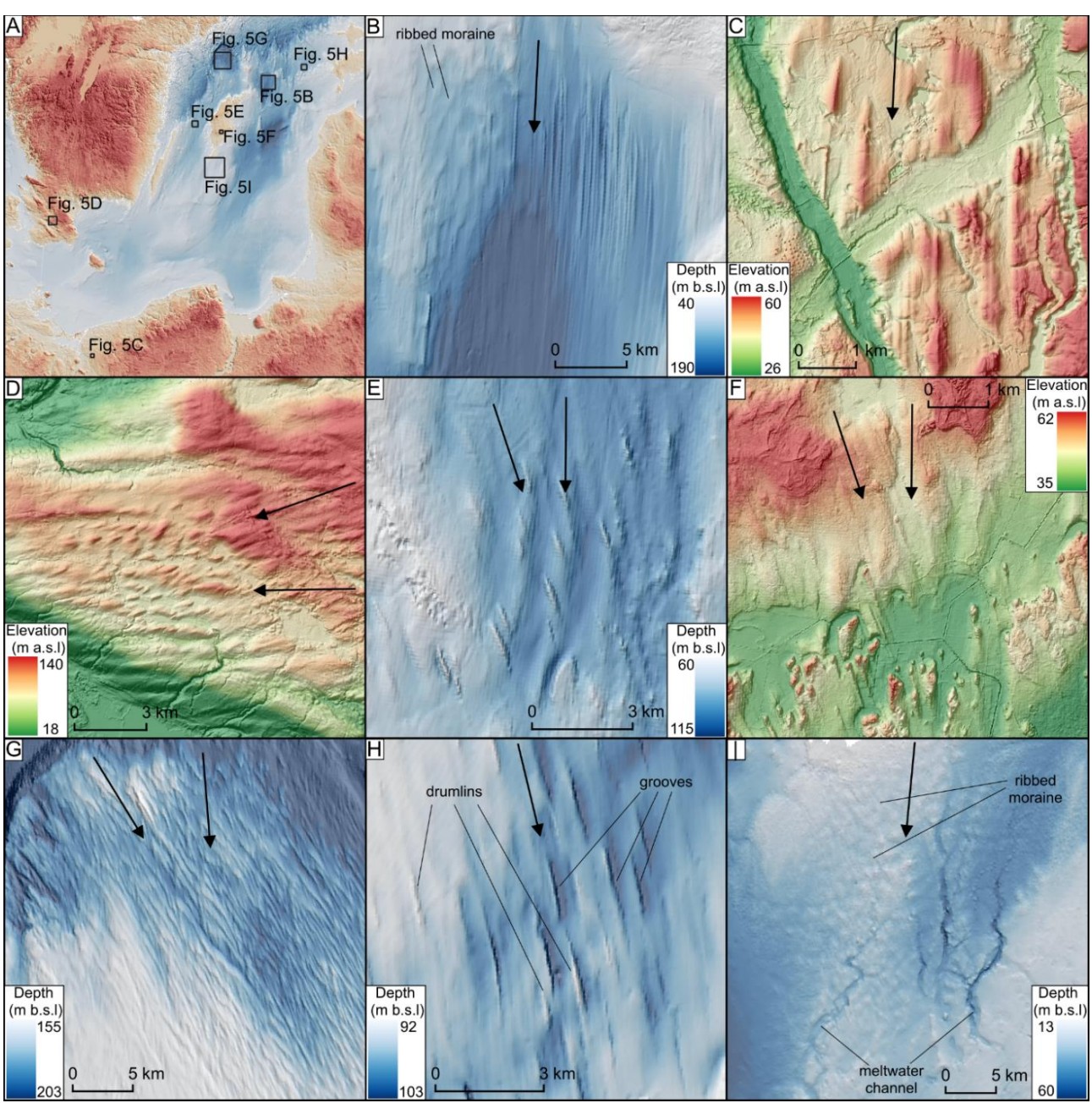

**Figure 5: Examples of glacial lineations, grooves, and ribbed moraines found in the study area. Ice flow direction is indicated by black arrows. The DEMs are based on EU-DEM and EMODnet if not stated otherwise (see Table 2). (A) Locations of landform examples. (B) MSGLs along Fårö Deep and ribbed moraines to the north-west. (C) Drumlins on the Polish coast. The DEM is provided by GUGIK (see Table 2). (D) MSGLs in Skåne, Sweden oriented E-W and cross-cut by NE-SW oriented drumlins. The DEM is provided by © Lantmäteriet. (E) Cross-cutting glacial lineations in the West Gotland Basin. (F) Crag-and-tails in southern Gotland, overprinting larger N-S oriented MSGLs. Note the moats (scour-marks) around the stoss sides of the crag-and-tails. The DEM is provided by © Lantmäteriet. (G) Cross-cutting glacial lineations in the vicinity of Landsort Deep. Notice that the NW-SE trending lineations toward East Gotland Basin are more elongated and better developed than the N-S trending lineations toward West Gotland Basin. This overlapping arrangement, seen here, is not common across the study area. (H) Glacial grooves identified in an area of basal till overlain by lacustrine and marine deposits (see Noormets and Flodén, 2002a) on the Silurian basement, between Klints. (I) Ridges south of Gotland Island oriented perpendicularly to inferred ice flow direction and interpreted as ribbed moraine.**

## 4.3 Flowsets and their relative age relationship

Figure 6 shows the 77 flowsets that were identified. The flowsets display a range of geometries with parallel lineation sets in deeper sections of the Baltic, and splaying geometries found across much of the study area from the terrestrial southern margin to the regions further north in the vicinity of the Klints. Flowset orientations vary substantially in the study region documenting ice flow directions to the E in the Eastern Baltic and to the NW in the western part of the study area. Overprinting (e.g., Figs 6B, C), on Gotland Island records 10 flowsets with orientations switching from N-S to NNE-SSW, through NE-SW, ENE-WSW, and NW-SE (Fig. 6C). Flowset relative timing was distinguishable in several parts of the study area by superposition, cross-cutting and smudging of landforms (Fig. 6). We suppose that such complex shifts in ice flow direction were likely common through the study area but are not always visible due to low resolution data.  Individual flowsets with no overprinting relationships (Fig. 6A) dominate in the eastern and the northernmost part of the study area.

## 4.4 Ice-contact and meltwater landforms

A range of ice-contact landforms were mapped (Fig. 7) including moraine ridges (e.g., Fig. 7B), ice-contact fans (e.g., Fig. 7E) and GZWs (Figs 7C, F, I). The most prominent ice-contact landforms are found in the central Baltic where they are up to 30 m high and 45 km long (Fig. 7E), with steep proximal and gentle distal slope. The only information on their sedimentological composition comes from limited seismo-acoustic profiles, which indicate that landforms are composed of till and glaciofluvial sediments (Uścinowicz 1999). The cross-profiles combined with the shallow water in the region during deglaciation indicate these are not grounding zone wedges but ice-contact fans. An ice-contact fan in vicinity of Słupsk Bank was interpreted by Uścinowicz (1999) as a glaciofluvial delta. The broad, up to 37 km long, 50 km wide, 34 m high, and only 13 m b.s.l.  ice-contact landform east of Öland Island (Figs 7E, I) has a steeper distal and gentle proximal slope with moraines superimposed on it. The size and shape resemble a grounding zone wedge (cf. Batchelor and Dowdeswell, 2015). Other ice-contact landforms to the north (along the Klints) typically have  steep distal slopes and wider and lower-angled proximal slopes (Figs 7C, F). They are composed of thick glaciofluvial sediments (up to 55 m; Noormets and Flodén, 2002a), and are up to 40 km long. These ice-contact landforms, along the Klints, are interpreted as grounding zone wedges (cf. Batchelor and Dowdeswell, 2015), although their limited amplitudes (from only 5 m) and local transitions to moraines may indicate that the free space beneath the ice shelf was shallow and floating occurred locally. Locally, moraines are visible on top of and next to

GZWs (e.g., Figs 7C, F, I). The two ice-contact landforms at Pomeranian Bay and Darss Sill – Falster-Rügen Plain (Figs 7D, 9B) are almost flat and 40 to 70 km long, composed mainly of glacial tills, Pleistocene sand and silt, and are covered by fine glaciolacustrine and marine sediments (see Lemke et al., 1994; Lampe et al., 2011; Kramarska, 1998) that resemble flat-topped ice-contact fans. Moraines represent a broad spectrum of sizes, from wide (up to 3500 m) and high (up to ca 20 m) to narrow

(less than 300 m), steep and up to 4 m high (Figs 7B, C, F). Their internal composition has been recorded in the southern Baltic (Fig. 9B), in the area near Darss Sill – Falster-Rügen Plain, where Jensen (1993) identified a series of small thrusted moraines composed of stratified clays overlain by glacial diamicton. A series of closely spaced moraines are also identified locally along Irbe Strait (Fig. 7C) and east of Öland Island (Fig. 7I) with 3–5 m amplitude and 300–400 width.

Cross-cutting and overprinting of moraines and ice-contact fans occur in the vicinity of the Danish islands, eastern Germany, and Słupsk Bank (Fig. 7D) in the central Baltic. Cross-cutting relationships such as those in the central Baltic, record changes in ice flow geometry during retreat. Here a distinct N to NW, NE to N and E-W flow geometry can inferred (Fig. 7E), with ice-contact landforms associated with the NE-N ice flow direction being the youngest.

Single- or multiple-crested eskers are common (n> 3000) in the west and southwest of the study area (e.g., Figs 7G, 9E) but are almost completely absent in the east (East Gotland Basin and eastern Baltic coast). Most of the eskers identified in this study occur onshore with only 2% of the total population located offshore. The longest and most complex eskers occur in southern Sweden, where they form dendritic networks (Dewald et al., 2022; Stroeven et al., 2016), and offshore from here, where eskers are up to ca 45 km long. The offshore eskers located in the vicinity of the Swedish coast conform to the splaying

pattern of the inland esker networks. Another group of SSW-oriented single-ridged offshore eskers, up to 45 km long, align with the streamlined bedforms in the deepest parts of the Baltic Sea. The two esker groups of opposing orientations are divided by the island of Gotland. Along the Klints, Noormets and Flóden (2002a) mapped several broad (up to 12 km wide) eskers However, we interpreted and mapped them as ice-contact landforms on the basis of the fan-shaped morphologies along at least 90 km (see the margin indicated by the black line starting at Gotska Sandön (SG) toward the east in Fig. 8), and their similarity

to moraines and GZWs presented in Fig. 7F. Eskers are not present in offshore regions of the southern Baltic, and are common but short (< 10 km in length) in the adjoining coastal areas of northern Germany and Poland (Frydrych, 2022). With the given DEM resolution, we did not detect any eskers at Adler Bank (Feldens et al., 2013). Here we detected only ridges oriented transverse to the ice flow direction that mimic small, 2-3 m high, recessional moraines (Fig. 4). The general distribution of better-developed esker networks on the shield compared to regions of sedimentary lithologies is consistent with observations

from the former Laurentide Ice Sheet in North America (Clark and Walder, 1994). We speculate that such small landforms might not be distinguishable in the poor-quality DEM available for the offshore areas or that eskers could be buried by postglacial deposits (Uścinowicz, 1999).

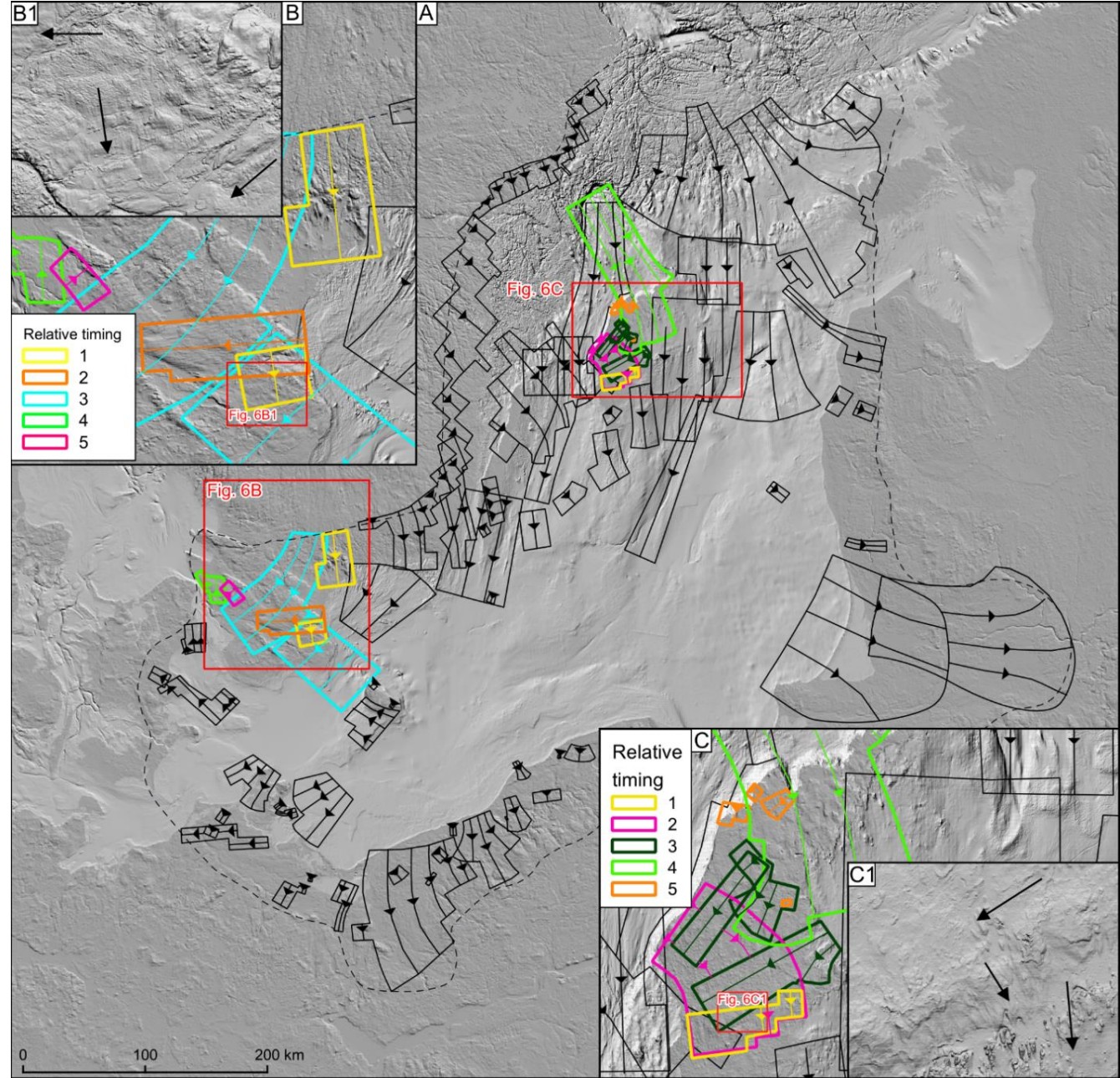

**Figure 6: (A) The flowsets defined for the study area (dashed line). Close-ups of flowsets in the area of (B) Skåne and (C) Gotland islands. Colours indicate their relative age based on overprinting relationships. Higher numbers in (B) and (C) are younger. The relative timing of green (4) and pink (5) flowsets in (B) is uncertain. The DEM is based on EU-DEM and EMODnet (see Table 2).**

Ice-marginal meltwater channels in the study are often associated with the lobate shape of the moraines located in their vicinity. They vary in size, from long (exceeding 100 km sections), wide (up to 5 km), and well-preserved channels (in, e.g., Germany, Poland, Kaliningrad) to short and fragmented (10-100 m wide) channel segments (Fig. 7H). The ice-marginal meltwater

channels along the Kaliningrad, Poland, and German coasts form semi-continuous drainage systems almost 400 km long. Offshore meltwater channels are generally restricted to the southern part of the study area; the widest examples here are up to 7 km in cross section (along the Polish coast).

Tunnel valleys are identified both onshore and offshore (Figs 7E, G, 9B, E), some up to 90 km in length. Tunnel valleys typically do not exceed 2 km in width, consistent with other examples from Scandinavia (cf. Jørgensen and Sandersen, 2006). They often occupy geological structures along the Swedish coast and the Klints, whereas in the southern sector they occur in unconsolidated sediments. Terrestrial examples are often associated with eskers, but only nine such associations are identified offshore.

Much of the sea bottom in the central and northern sector of the Baltic Sea are covered with iceberg ploughmarks (Figs 4, 7F). They have an identifiable U- to V-shaped cross-profile with thalwegs that are typically curvilinear to straight. The ploughmarks rarely exceed a depth of two meters and their length varies between 45 m and 20 km, consistent with other studies on ploughmarks in the region (Dorokhov et al. 2018; Karpin et al., 2021). However, in some areas the resolution of the DEM hampered ploughmark detection, e.g., the ~ 220 km$^2$ field detected by Dorokhov et al. (2018) in central Baltic in the vicinity of our large, ~ 16,970 km$^2$ field. Orientations vary from perpendicular to the inferred ice margins to chaotic with ploughmarks cross-cutting each other. No ploughmarks are identified in the southernmost part of the study area (Fig. 4), down-ice from Adler and Odra banks, either because of the low DEM resolution or due to shallow water depth.

### 4.5 Ice margin positions

To reconstruct the character of ice margins and the pattern of deglaciation we used the suite of mapped landforms and our glaciological understanding of how they formed to infer major ice margin retreat geometries (Fig. 8). The margin configuration along the terrestrial southern sector spanning Denmark, Germany, Poland, Kaliningrad region and Lithuania comprised a series of lobes as evidenced by meltwater channels, eskers and moraines. The spacing between the identified ice margin positions varies, with the southern sector generally having a closer spacing of about 10 to 40 km. Ice retreat down a reverse slope is reflected in the presence of long (almost 400 km) ice marginal channel systems that are well-preserved along the German, Polish, and Kaliningrad coasts. Offshore, ice margin positions are mainly identified from ice-contact landforms. Along the eastern Baltic ice margin positions typically have a 40 – 80 km spacing, with 180 km between the central Baltic ice-contact landforms and the Devonian Klint (Fig. 1), and 120 km between the Silurian Klint and Åland Sill. In the western Baltic the spacing is similar, up to ca 80 km, with a larger gap of ca 100 km east of Öland Island. It is probable that uniform and better-quality data for the whole Baltic area would permit the detection of more closely spaced landforms.

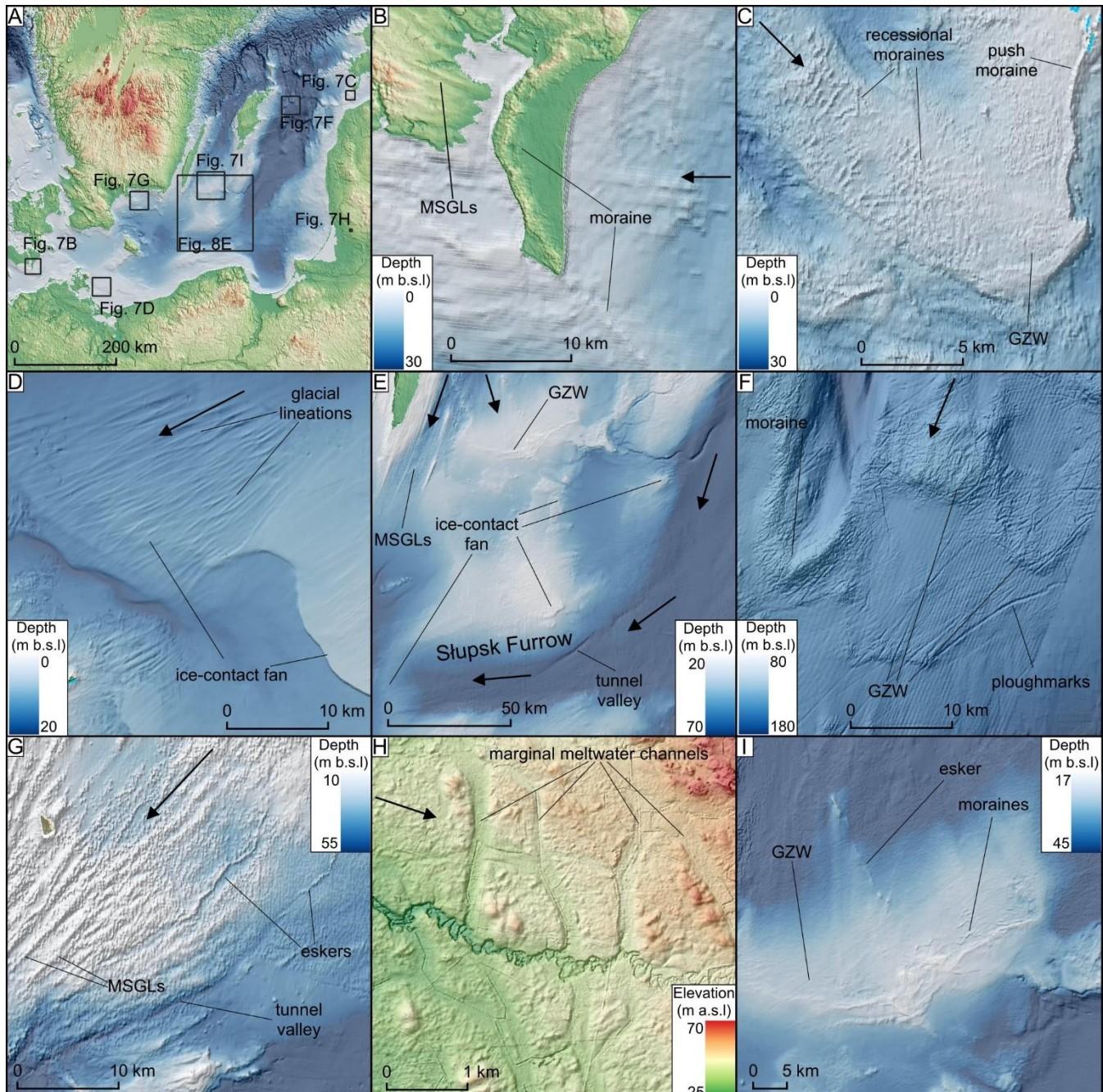

**Figure 7: Examples of ice marginal landforms. Ice flow direction is indicated by black arrows. The DEMs are based on EU-DEM and EMODnet if not stated otherwise (see Table 2). (A) Location of landform examples. (B) Example of moraine, onshore and offshore, in Denmark. The onshore DEM is provided by Styrelsen for dataforsyning og effektivisering (see Table 2). (C) Sequence of small, closely spaced moraines at Irbe Strait on the top of grounding zone wedge with a push moraine visible to the east. (D) Ice-contact fan overprinted by enigmatic lineations, possibly of glacial origin, in the shallow waters of Pomeranian Bay. The sharp banks truncating the lineations are thought to form due to drainage of a local ice-dammed lake (see Uścinowicz, 1999). (E) The central**

**Baltic, near the Słupsk Furrow, is occupied by the most prominent ice-contact landforms (ice-contact fans) in the study area. Tunnel valleys are oriented towards the westernmost ice-contact fan. East of Öland Island a grounding zone wedge is visible (see close-up in (I)). (F) Glaciofluvial deposits down-ice from a pinning point (Noormets and Flodén, 2002b, a) comprising GZWs and moraine, covered by ploughmarks. (G) Eskers and an associated tunnel valley system in Hanö Bay, parallel to MSGLs. (H) Examples of marginal meltwater channels in Latvia. The DEM is provided by Latvijas Ģeotelpiskās informācijas aģentūra (see Table 2). (I) The GZW east of Öland Island with a series of small moraines and an esker in the central part. Notice that the lobe is in contact with the ice-contact fan to the east; for a zoom out look at (E).**

In some offshore localities the ice margin separates into two lobes, with a clear interlobate (suture) zones (Fig. 8). In Hanö Bay and east of Öland Island, the ice margin forms two 55-80 km wide lobes, each with eskers oriented perpendicular to the ice-contact landforms (e.g., Figs 7I, 8 9E). In the central part of the study area, separate arcuate ice margin positions are reconstructed east and west of the SW-NE oriented ridge of Gotland Island and its offshore southern continuation (Fig. 8). The marginal pattern becomes more lobate to the north of the study area, with a clear interlobate (suture) zone along Gotska Sandön (GS in Fig. 8).

## 5 Discussion

Our landform mapping of the previously mostly unexplored Baltic Basin (0.3 million km$^2$) (although see also Greenwood et al., 2023) provides new information on the dynamics of the southern margin of the SIS during the last deglaciation. While our paper was in the reviewing stage a similar study on glacial landforms of the Baltic was published (Greenwood et al., 2023) and we suggest that interested readers should consult both sources noting that they were entirely independent investigations. Our study presents the glacial geomorphological mapping for the whole Baltic Proper, and adjacent coastal zone, but is of lower resolution for offshore areas, whereas Greenwood et al. (2023) is based on available patches of higher resolution data in the Baltic but does not include coastal zones and presents fewer details for the central and southern Baltic. Reassuringly, there is a large amount of similarity between the two studies, especially for the general offshore glacial lineation pattern, eskers and ploughmark positions. The largest differences in mapping concentrate on Gotland Island and south of Gotland Island where the higher resolution data used by Greenwood et al. (2023) allowed for detection of more ribbed moraines and eskers.

Our mapping helps to fill the gap in reconstructions of the SIS over the Baltic Sea (e.g., Boulton et al., 2001; Stroeven et al., 2016; Tylmann and Uścinowicz, 2022) that previously had to rely on inferences and extrapolations from the adjacent terrestrial evidence. Below we use the reconstructed flowsets and ice margin positions to address the research questions raised in the introduction.

### 5.1 Was there a Baltic-wide ice stream?

The existence of a large Baltic-wide (300 km) ice stream at the local LGM, feeding ice to the southern margin and westwards to Denmark (e.g., Holmlund and Fastook, 1995; see Fig. 2; Stephan, 2001) is largely based on glaciological inferences stemming from the broad-scale morphology of the ice sheet bed and its lithological properties, rather than on a landform-

defined footprint of the ice stream. We do not identify a landform 'footprint' of simultaneous fast ice flow spanning the Baltic depression; the landforms in the central Baltic appear to record more fragmented corridors of fast ice flow consistent with later stages of deglaciation and smaller ice stream widths (typically 30 to 60 km, up to 95 km; Figs 6, 9C). Such widths are more typical of ice streams found elsewhere (e.g., Gandy et al., 2019; Livingstone et al., 2012; Margold et al., 2015; Stokes, 2018). This is indicated by the presence of distinct flowsets located in the vicinity of Gotland Island (Fig. 6) and multiple lobes

indicated by the marginal retreat geometry in the central Baltic (Fig. 8). We interpret the flowsets confined to the depressions and flanking slopes either side of Gotland (Figs 6, 9C) and consisting of highly parallel MSGLs (Fig. 4) as the trunks of a number of narrow (30 - 60 km) palaeo-ice streams that operated during deglaciation behind a back-stepping ice margin. Whilst ice must have earlier traversed the Baltic to reach the maximum southern extent of the last SIS, as evidenced by the provenance of glacigenic sediments in the southern study area (e.g., Kjær et al., 2003; Woźniak and Czubla, 2015) we do not find any

landform evidence in the Baltic depression that we could interpret as originating from this stage. Landforms from the maximum phase of glaciation may have been erased, buried or have yet to be found.

Some ice sheet modelling experiments (e.g., Patton et al., 2017; Patton et al., 2016) reproduce a Baltic-wide area of fast flow (an ice stream) through much of the growth and decay of the ice sheet. We suggest that this fast flow zone is a response to the

prescription of a soft substrate in the Baltic in contrast to the hard-bedded surroundings (Patton et al., 2016, 2017). Modern observations of, for example, the Siple Coast ice streams in Antarctica (Catania et al., 2012), and numerical modelling experiments (Fowler and Johnson, 1996; Hindmarsh, 2009; Payne, 1998), suggest that narrow corridors of fast ice flow can exist even in soft-bedded regions of little relief; this is, in fact, what Bennett (2003) defined as 'pure ice streams'. We therefore suggest that some width-limiting processes on ice stream formation have yet to be sufficiently added into numerical ice sheet

models. Gandy et al. (2019) have made progress with such work for the British-Irish Ice Sheet where they explored the ingredients required in their modelling experiments to simulate ice streams in the correct places and with appropriate spacing and widths. They used the BISICLES higher order model, which as well as having variable cell-size resolution, which helps with apportioning ice steaming, also incorporates approximations of membrane stresses which are necessary for yielding accurate ice streams.

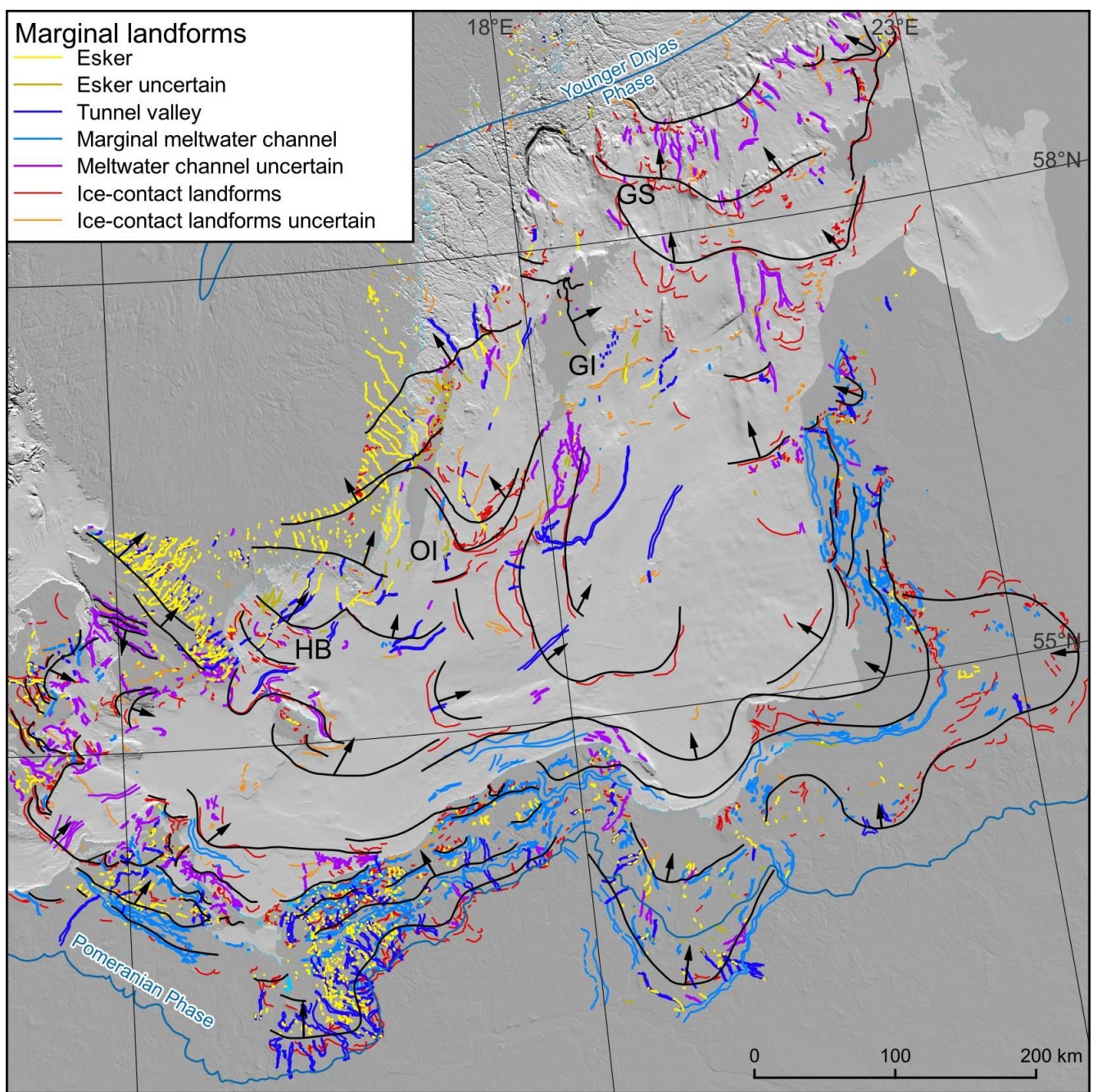

**Figure 8: Broad overview of possible ice marginal retreat geometries (black lines with arrows showing the direction of retreat) inferred from the new mapping of landforms. Gaps in the bathymetric data and the known complexity of flowsets and flow-switching through time mean that the ice sheet retreat was likely more complex than illustrated here. Notice, the highly lobate nature of the ice margin to the southwest and southeast and some offshore examples of interlobate (suture) zones where the ice margin separates into two lobes: at Hanö Bay (HB) and over Öland Island (OI); along Gotland Island (GI) and Gotska Sandön (GS). The DEM is based on EU-DEM and EMODnet (see Table 2).**

**5.2 What was the character of ice margin retreat though the Baltic depression?**

Rather than congruent Baltic-wide ice streaming, flowsets can generally be incorporated into a time-transgressive deglacial retreat sequence that in some cases conforms with the pattern of reconstructed ice margins (Fig. 8). We find that the ice margin geometry changes significantly during retreat through the Baltic (Figs 9B, C). In the southeast and southwest, lobate moraines with splaying geometries, flanked by marginal meltwater channels (Figs 7B, 8, 9B), are common, resembling lobate land terminating margins documented elsewhere on the soft-bedded (unconsolidated sediments) southern and eastern margins of

the SIS (e.g., Kalm, 2012; Szuman et al., 2021), and Laurentide Ice Sheet (Margold et al., 2015). Ice marginal signatures in the southern sector often comprise overprinted lobes interpreted to arise from ice-margin oscillations and readvances (Figs 6, 9B) along with switches in flow orientation and changing lobe positions during overall retreat (cf. Pedersen, 2000; Kjær et al., 2003; Gehrmann and Hardig, 2018).

The presence of ice marginal meltwater channels in Denmark, Germany, Poland, Kaliningrad and Lithuania, and in the present-day coastal zone indicates that the marginal environment in this phase of retreat was terrestrial (Fig. 4) (cf. Uścinowicz, 1999). Flat-topped ice-contact fans, in the southernmost sector (i.e., Darss Sill - Falster–Rügen Plain and Pomeranian Bay; Figs 7D, 9B) indicate the position of ice margins (Lemke et al., 1995 and the references therein) and deposition of fine sediments in a shallow glaciolacustrine environment once the ice margin stepped back north of the present-day coast. While these ice-contact

landforms resemble the shape of grounding zone wedges (cf. Batchelor and Dowdeswell, 2015), the water depth, only up to 30 m along the coast (Uścinowicz, 2003), is considered too shallow for ice shelf development. However, we note here the strong resemblance of the features we map to the Pas Moraine in Manitoba, which has recently been re-interpreted as a grounding zone wedge formed in Glacial Lake Agassiz (Gauthier et al., 2022).

In the southern part of study area, numerous ice dammed lakes have been reconstructed in and adjacent to the Baltic Depression from sedimentological investigations (Houmark-Nielsen and Kjær, 2003; Larsen et al., 2009) and the identification of palaeo-shorelines (Uścinowicz, 1999). The largest of these lakes is the Baltic Ice Lake (Björck 1995; Uścinowicz, 2006; Rosentau et al., 2017). During the retreat of the SIS through the Baltic Depression, the ice margin may have terminated on land, or been grounded in an ice dammed lake, and we suggest that the varying geomorphologies noted above likely reflect these terminal

environments. During the early stage of the Baltic Ice Lake the area south of Słupsk Furrow, Odra and Adler banks were elevated above the lake water (Uścinowicz, 1999). North of prominent central Baltic ice-contact landforms, east of Öland Island, the conditions were suitable for GZW (Fig. 7I) formation. Further north, parallel MSGLs are aligned within the deeper depressions (Fig. 9C). Here specific MSGL sets cannot be conclusively linked to ice marginal landforms. Given their highly parallel nature, alignment with the orientation of depressions that are up to 240 m deep and overprinting by iceberg

ploughmarks, we suggest these landforms are indicative of a water-terminating ice margin with icebergs calving either directly at the ice front or from an ice shelf that might have existed over the deeper sections of the Baltic Ice Lake. The change from the ploughmark-free southern sector (Fig. 4) to numerous ploughmarks in the deeper (up to 220 m b.s.l.) northern sector likely

reflects a transition from land-terminating or shallow lacustrine margin to a deeper aqueous calving margin. Similar parallel corridors of MSGLs (i.e., not splaying) have been found at former marine-terminating margins of the Antarctic Ice Sheet (Livingstone et al., 2012). This is consistent with Noormets and Flodén (2002b), who suggested an ice thickness of 180 m at the grounding line, with a floating ice shelf over Fårö Deep (see Fig. 1 for location). The distal slopes of elevated sediments at pinning points especially along the Klints are covered with ploughmarks (Fig. 7F; see also Karpin et al., 2021), indicating either a long still-stand or that the calving processes were intensive, and the icebergs were smaller than the full depth of ice, allowing them to pass through topographic barriers.

## 5.3 What was the role of bedrock structures in controlling stepped retreat?

Glaciolacustrine sediments are widespread in the Baltic (Boulton and Jones, 1979; Boulton et al., 1985; Noormets and Flodén, 2002b, a; Uścinowicz, 1999) however, there are locations where bedrock is found to outcrop at the seafloor including at the Island of Bornholm and the Klints in the northern Baltic (Fig. 1; Tuuling and Flodén, 2016). These outcrops are generally the result of more resistant geological units producing topographic protrusions. The topographic relief associated with the Klints is associated with ice-contact landforms (Figs 8, 9D) including outwash fans (Noormets and Flodén, 2002b), GZWs and moraines (e.g., Fig. 7F) and tunnel valleys (Fig. 4). Lineations further north splay according to the Klints' position (Fig. 9D) and can be incorporated into a stepped pattern of retreat. The quasi-regular pattern of ice margin positions (Fig. 8) gives an impression of stepped recession. In our interpretation, stepped retreat could be related to the presence of pinning points formed by the Klints and other topographic/bathymetric features (cf. Boyce et al., 2017; Noormets and Flodén, 2002b). In the study area, the presence of ice-contact landforms along the higher elevated Darss Sill – Falster Rügen Plain and Odra Bank (Figs 7D, 9B), in the central Baltic (Fig. 7E; near Devonian Klint from Fig. 1), along the Klints (Figs 7F, 9D, F; see also Noormets and Flodén, 2002b), north of Bornholm Island (Fig. 9E) is consistently associated with these geological structures (see Kramarska 1998; Noormets and Flóden, 2002a; Jensen et al., 2016). The Darss Sill and Odra Bank have hard bedrock elevated by ca 30 to 50 m compared to the Arkona and Bornholm basins (Jensen, 1995; Tomczak, 1995; Flodén et al., 1996). In East Gotland Basin the ice was steered and pinned by the higher elevated Devonian Klint to the east (see Fig. 1 where the limits of East Gotland Basin follow the geology) and the extension of the Gotland Island ridge to the west. Thus, significant (the most prominent zones of sediment accumulation along Baltic depression) accumulation of unconsolidated sediments along geological steps indicates the importance of pinning points in ice stabilisation (e.g., Favier et al., 2016; Still and Hulbe, 2021). The only prominent high on the Baltic bed where we do not map accumulations of glacigenic sediments is Åland Sill, possibly because of patchier sediment availability over the crystalline bedrock.

## 5.4 Was there an ice flow interaction between northern (Bothnian Basin) and north western (Swedish) ice sources?

Past ice-sheet-scale investigations have inferred an ice lobe with a gently arcuate ice front spanning the width of the Baltic depression during northwards retreat, implying that ice was steered through the Gulf of Bothnia (Boulton et al., 2001). We confirm this Bothnian ice source from the north, but also identify a Swedish ice source with a ~NW-SE orientation based on

our ice marginal and flowset geometries (Figs 6, 8, 9C). These landforms record the interplay between two ice sources, with the high-resolution DEM from the islands of Gotland and northern Öland assisting in the identification of five independent ice flow directions (Figs 6C, 9C). The zig-zag-like pattern of flowsets with ice flowing from the NW-SE (Swedish source) and N-

530 S to NE-SW (Bothnian source) indicate alternating activity of both ice sources. Our mapping indicates that the two ice lobes along West and East Gotland basins were in contact (Figs 7E, 9E) in the earlier stages of deglaciation but with a later switch to dominance of the eastern lobe. The suture between the two lobes was located along Gotland Island (Figs 8, E) and extended towards Gotska Sandön in the north and ca 75 km to the south. The multi-channelled meltwater system (Fig. 5I), widespread accumulation of glacifluvial sediments (here expressed as ice-contact fans; see also Noormets and Flodén (2002b) for evidence

of glacifluvial sediments along Gotska Sandön) and a concentration of MSGLs on both sides of Gotland Island are typical characteristics of suture zones (e.g., Punkari, 1997; Evans et al., 2018). We also identify more localised ice flow splitting north of the island of Gotland, indicated by diverging streamlined bedforms (N-S and more NW-SE) in Landsort Deep (Figs 6A, 9F).

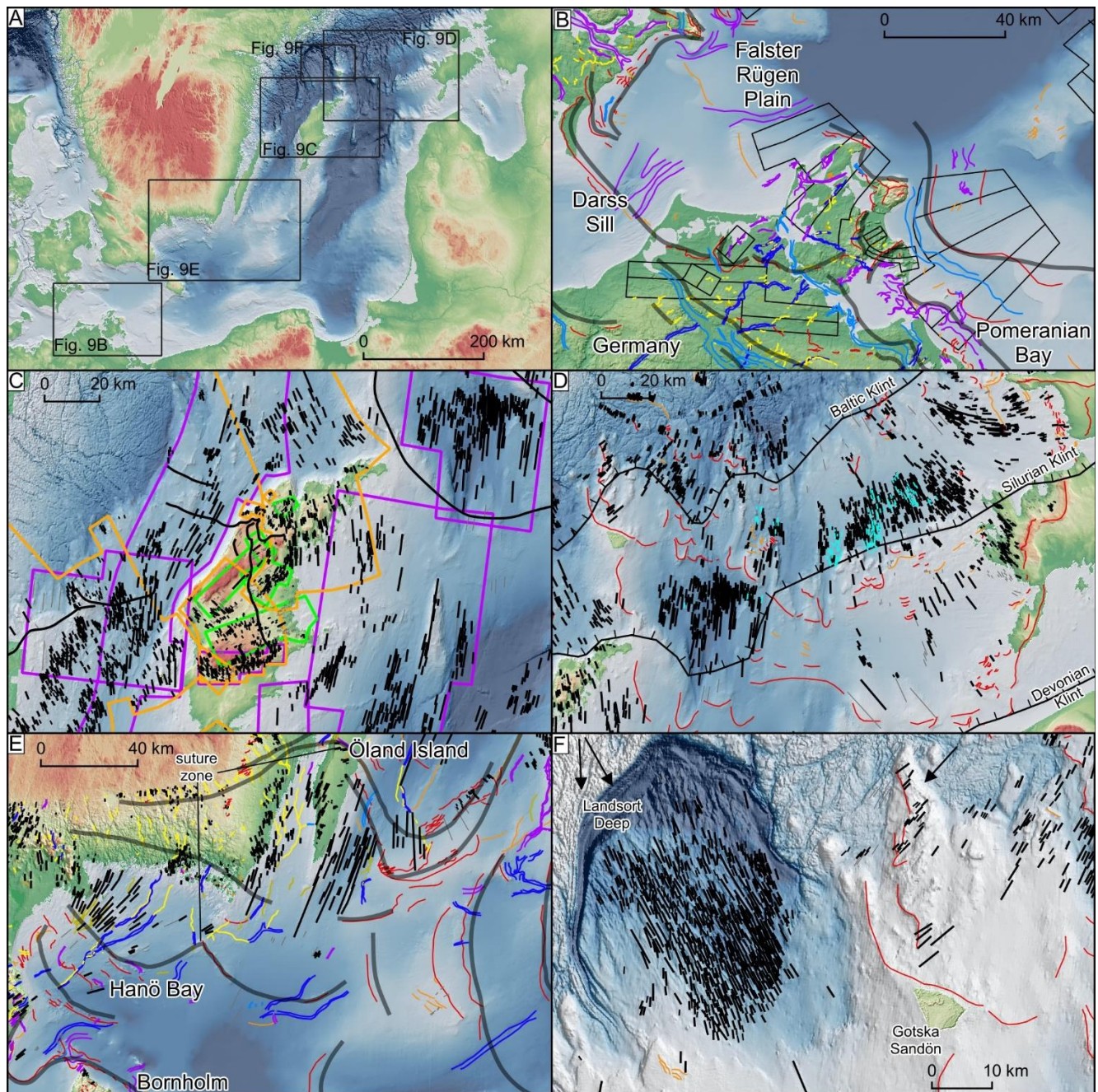

Figure 9: (A) Location of panels B-F. The DEMs (A-F) are based on EU-DEM and EMODnet if not stated otherwise (see Table 2). (B) Lobate ice margin positions along the German coast, with a corresponding lobate pattern of flowsets and onshore-offshore ice-contact landforms (the legend is indicated in Fig. 4), (C) highly parallel MSGLs in the bathymetric depressions (purple polygons). The parallel MSGLs indicate ice stream widths of roughly 30-60 km, with a maximum up to 95 km. Notice that splaying lineation patterns are rare in this location. The orientation of ice flow within the classified flowsets is indicated by colours: violet for N-S to NNE-SSW oriented (possibly the oldest), orange for NW-SE (younger), green for NE-SW to ENE-WSW (the youngest). The flowset

pattern underlines the interplay between ice margins with Swedish-oriented and Bothnian-oriented ice sources. The onshore DEM is provided by © Lantmäteriet (see Table 2). (D) ice-margins associated with topographic highs formed by the Klints (black lines). Notice the highly lobate shape of ice-contact landforms that fits to the position of the Klints. Glacial lineations have splaying patterns, especially down-ice from the Baltic Klint. The onshore DEM was provided by Estonian Land Board 2021 (see Table 2). (E) Suture zones in Hanö Bay, Öland Island and the central Baltic. Note ice splitting into two lobes, indicated by the pattern of eskers (yellow) and tunnel valleys (blue), and by the positions of ice-contact landforms (red), especially well visible to the east (a swarm of 'red' margins; here the East and West Gotland lobes are in contact). The onshore DEM is provided by © Lantmäteriet (see Table 2). (F) Glacial lineations in vicinity of Landsort Deep and Gotska Sandön indicate ice flow divergence towards the SE and S from the Landsort Deep and SW towards Gotska Sandön.

## 6 Conclusion

A Baltic-wide glacial landform-based map is presented, filling in a geographical gap in the record that has been speculated about by palaeoglaciologists for over a century. The glacial landforms we map are interpreted as primarily recording phases of ice retreat through the Baltic, rather than an extensive landform record related to the maximum ice extent phase. Instead of an often interpreted or modelled accelerated ice flow zone spanning the Baltic depression (300 km), we provide landform evidence for narrower corridors of fast ice flow that we interpret as distinct and smaller ice streams (widths of 30 to 60 km, up to 95 km), consistent with later stages of deglaciation rather than the maximum stage. In the central Baltic, assemblages of landforms resemble lobate ice margins typical of terrestrial-style landsystems, whereas locally within the deeper bathymetric depressions landform assemblages more typical of water-terminating ice margins are found. In these latter cases ploughmarks suggest significant ice evacuation by calving. Episodic rather than steady ice retreat is inferred based on ice margin positions associated with exposed bedrock structures that likely acted as pinning points. Where previous ice sheet-scale investigations inferred a single ice source, our mapping identifies flow and ice marginal geometries from both Swedish and north Bothnian sources. We anticipate our landform mapping and interpretations may be used as a framework for more detailed empirical studies by identifying targets to acquire high resolution bathymetry and sediment cores and also for comparison with numerical ice sheet modelling.

**Funding sources**

This research has been supported by the Polish National Science Centre (NCN) (grant no. 2015/17/D/ST10/01975). CDC and CRD were supported by the European Research Council, H2020 (PalGlac grant no. 787263).

**Author contributions**

IS and JZK conceived the project. IS led the project and wrote with CRD the initial version of the manuscript, which was subsequently improved by the contribution of all co-authors. IS, JZK mapped the landforms. All co-authors contributed to the interpretation of problematic areas.

**Competing interests**

The authors declare that they have no conflict of interest.

**Acknowledgements**

We are grateful to two anonymous reviewers and Sarah Greenwood for their valuable comments and Marek Ewertowski for constructive discussions.

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
