# Peer review of "Reconstructing dynamics of the Baltic Ice Stream Complex during deglaciation of the Last Scandinavian Ice Sheet"

_The Cryosphere, 2023_

## Referee Comment (RC1)

**Evaluation of the role of the Baltic depression during deglaciation of the last Scandinavian Ice Sheet; a landform-driven investigation**

Izabela Szuman1, Jakub Z. Kalita1, Christiaan R. Diemont2, Stephen J. Livingstone2, Chris D. Clark2, Martin Margold3

*1 Department of Geomorphology, Adam Mickiewicz University, Poznań, 5 62-712, Poland*
*2 Department of Geography, Sheffield University, Sheffield, S10 2NT, United Kingdom of Great Britain - England, Scotland,Wales*
*3 Department of Physical Geography and Geoecology, Charles University, Prague, 128 43, Czech Republic*

**General comments:**

This study is an ambitious and laborious task mostly based on combination of bathymetric data sets of different resolution supported by data from earlier offshore studies and "onshore literature". Approaches to utilize increasing amount of high-resolution bathymetric data are crucial to promote scientific discussion on the distribution of submarine glaciogenic features and the influence of sea-bed bedrock/topography on glacial dynamics, especially in the case of streaming ice. The manuscript will add to progress in numerical modelling of the Scandinavian or Fennoscandian ice sheet.

In general, structure and proper length of the manuscript make it easy to crasp. However, the major problem with the manuscript is the quality of the mapping procedure with some likely misinterpretations or lack of alternative explanations in the presented data, as also indicated by *Greenwood et al. in their interactive comments. Moreover, like commented by Greenwood et al., the use of geological maps for coastal periphery-offshore interpretations would have increased the quality of the methodological approach.* Uncertain observations on maps should be better presented/visualized and reasoning/data behind some interpretations, regarding f. ex. moraine landforms, should be better explained. It is important that the results and "dynamic history of the Baltic sector of the SIS" presented in Figure 8 are based on "solid" ground. Nevertheless, it is likely that some mistakes or misinterpretations remain in these kinds of maps based on large data sets with varying resolution. To sum up, the results presented in the manuscript do not convincingly enough support the interpretations made. I think the manuscript should be published after major revision, answering to above-mentioned main concerns (as also interactively commented by Greenwood et al.), and including re-evaluation of some key observations or interpretations pointed out in the specific comments.

**Specific comments:**

Title: the title is ok but not very "selling" in terms of the glacial dynamics or methodology used.

L_16 (Abstract): … "that might have resulted in rapid collapse of this ice sheet sector." <> possible rapid collapse of the ice sheet is not discussed in the main text – it should be removed from the abstract or discussed!

L_37: I would not use the term conduit here, very broad ´channel´ is simple enough.

L_38: it would be relevant to include information/references on the shifting ice-divide along the Scandes mountains and its probable influence on the ice dynamics!

L_39: you should maybe use more accurate definition here (unconsolidated). There is also different behavior/effect of soft sediments, depending on their composition and grainsize distribution.

L_41-42: "impeding water evacuation from the ice margin" <> not from the ice-margin but from the proglacial basin?

L_53: I think this research question is not really relevant here, the Baltic depression surely had an overall, prominent role for LGM position but related details are more poorly known + in the end you also answer first to the second question which apparently is the main issue.

Table 1: Feldens et al., Dorokhov et al., Schäfer et al., Jensen, All et al. and Jakobsson et al. 2016, 2020 are not mentioned/cited in the text. If these studies are important for the landform interpretations, they should be mentioned!

(Figure 2: "No submarine evidence was available at the time Boulton et al. (2001) drew the figure and it is a purpose of this paper to seek information about the footprint of this ice stream and its likely width." <> this sentence is simply well put and would be good to replace it into the introduction as well?)

L_125-130: "Our motivation for mapping the coastal periphery of the Baltic is twofold:

1. It eases integration with the much-studied onshore landform record,
2. To provide a high-resolution verification for ice-flow patterns and ice-marginal retreat patterns mapped from the lower quality and resolution bathymetric dataset.
<> would it be better to place this text into Methods?

L_142: zones of soft sediment (similar consolidation or properties?)

L_144: could these fan-shaped belts be indicated on a map?

L_146: soft sediments has been found ... <> does this mean that elsewhere in the basin there are no soft sediments?

L_151: cross-checking <> this is a bit unclear way to express this – where there always hydroacoustic or seismic data available for artifact recognition? – somehow related to works presented in Table 1?

L_157: Mapping was carried out ... <> in this case the use of geological maps to verify interpretations (where possible) would have been necessary. It is important to include information from geological maps to methods as it likely strengthens the interpretations along the coastline-offshore regions. *Also Greenwood et al. raise this point in their interactive comments!*

Table 2: for the future work, in case there are changes to databases, it would be good to indicate the date when the data was collected (f.ex. 02/2023)

L_168: you mean glaciolacustrine wedges, please explain the term shortly when mentioned for the first time.

L_170: tunnel valleys <> do these include possible (braided) subglacial meltwater routes or corridors not strictly classified as tunnel valleys? See possible route in Fig. 3B just below the text: good quality DEM?

L_179-180: "identifying larger features in these regions including moraines, ice marginal channels, and ice marginal deposits was often still possible" <> what are these interpretations based on? – any references/supporting earlier data or just interpretation based on ...?

Figure 3: 3C) offshore area west of the Gdansk Bay has been mapped to show marginal meltwater channels (Fig 4) <> why not longshore structures (bars & troughs) related to a large spit-platform?

L_200: remove likely

Figure 4: would it be possible to explain in the text why there are so little landforms within the offshore areas located SW and NE from the Landsort Deep?

L_232: Fig. 5B <> where are ribbed moraines in this figure? - wrong fig?

Figure 5G: these landforms are not presented on a map!

L_261: (Fig. 7E) <> I think that this ice-marginal moraine interpretation is not well justified - any reference from earlier marine investigations or reasoning behind the interpretation?

L_267: (Fig. 7D) <> in this figure?

L_268: distinct (?) + inferred (Fig. 7E), with <> I can´t see this in Fig. 7E

Figure 6B: the leftmost flowset probably has a wrong arrow direction

L_301: geological structures <> can you be a bit more precise?

L_308: "Iceberg pits are identified in one locality" <> any reference for the interpretation, what is this interpretation based on?

Figure 7D: I´m not convinced that these are glacial lineations! - too non-uniform + winding/overlapping - for me these landforms rather resemble submarine erosional/depositional features by bottom currents!

Figure 7F: why are these interpreted as moraines, any earlier supporting data?

Figure 8: although there likely will remain some erroneous interpretations/details of landforms in this massive data set, this figure (black lines) should stand strong on hard evidence or if black lines are based on uncertain or extrapolated data, they should be marked with f.ex. dashed lines

L_403: 9B <> not shown in the figure

L_405-406: "An example of this is in Pomeranian Bay where a glaciolacustrine fan is overprinted by long cross-cutting MSGLs, indicating a subaquatic origin." <> sorry, but I do not understand this sentence (or reasoning), and how does it relate to previous sentence. MSGLs refer to ice flow over the wedges BUT I think these MSGLs are not glacial features at all but instead submarine current structures!

L_416: prominent moraines <> based on what? - any other supporting data than DEM interpretation?

L_427: (Figs 8F, I; <> wrong figure reference

L_436: outwash fans <> these not mapped in the present study?

L_437: moraine interpretation based on what?

L_450-452: "We confirm this Bothnian ice source from the north, but also identify a Swedish ice source with a ~NW-SE orientation based on our ice marginal and flowset geometries (Figs 6, 8, 9C)." <> just to check this - are you absolutely sure that you are the first ones to report this Swedish ice source in this area?

L_457-458: Bedforms within the Landsort Deep are not presented on map in Fig 4! The Deep is an interesting and outstanding submarine geological feature, so it would be good to discuss its origin (+ references) - could there be any alternative explanation for the features interpreted here as glacial lineations?

**Technical corrections:**

L_22: MSGL should be opened when used for the first time and abbreviation used later in L_25

L_26: rather than

L_49: , encompassing

Figure 1: index map is of slightly poor resolution and the scale bar is missing from the main Figure 1B.

Figure caption: Svendsen et al. 2004 not in the references list!

Figure 1D: could the two troughs of Gotland Island be indicated on the NW-SE profile?

L_74: 1.1 <> should this rather be 2 Background?

L_77 and 86: northwestward (in my opinion)

L_95: , indicating

L_134: northwestern

L_139: sediment … the northern part

Figure 3: 3A) Gotland is situated at left so west Gotland basin is wrongly placed, 3B+D) conglometare? Figure caption says conglomerate.

L_194: Commonly, it is not good to start the sentence with number

L_211: exhibit high

Figure 4: use of colours for the lines is problematic even for slightly red-green blind reader (like me). In addition to colours also different line styles could be used to clarify visualization.

Figure 4: "Light and dark grey indicate onshore and offshore areas respectively" <> I think this should be the other way around

L_248: 75 flowsets

L_260: , including

L_315: lobes as

Figure 8: Again, in addition to colours, different line styles would improve readability! f.ex. uncertain observations could be marked with dashed lines? In southern Sweden there is one black line containing two different retreat directions? + Scale bar is missing!

---

## Community Comment (CC2)

Exploitation of newly available terrain datasets in the Baltic region is undoubtedly welcome, since the offshore sector was both dynamically important to the last ice sheet and highly under-researched. However, we identify a number of concerning problems with the work presented here, notably the quality and rigour of the landform mapping. We comment only on internal issues in the manuscript. We do not discuss interpretations guided by any data not available to these authors, only what is presented as figures or text in the manuscript.

1. Mapping rigour

- We find numerous examples of erroneous landform interpretation: bedrock ridges (with little/no sediment cover), dolerite dykes, aeolian dunes, marine current bedforms, estuarine banks, among others, mapped erroneously as glacial landforms.
- We find numerous examples of inconsistent mapping choices: neighbouring landforms of similar appearance mapped or unmapped.
- Landforms visualised in figures have not actually been mapped (e.g. Landsort Deep, Fig. 5G, 9F).
- The approach to moraine mapping offshore is very unclear: it appears that all 'bumps' not considered a lineation, rib or esker are recorded as moraines, with little/no discussion of or motivation for the interpretation. This is problematic for the discussion of the retreat pattern (e.g. Section 4.5).
- Consultation of geological maps (easily and publicly available) would have avoided many mapping errors. (Statements like L264-5 - lack of information on composition - are false.)

We provide a selection of examples of mapping errors in the Figure, below, and note that these examples are just a few of those we encountered.

Furthermore, with regards to mapping methods:
- There are three versions of the tiled EMODnet DTM product (2018, 2020, 2022); the authors do not state which version was used. These versions have noticeable differences in landform visibility stemming from differences in input datasets and (re-) gridding results. Comparing with visually obvious gridding artefacts, the base topography in their Fig 4 appears to be the 2018 version - did they use more recent data too?
- L154: "Artifacts are common in the dataset and where these occurred, cross-checking in the data from hydroacoustic surveys and seismic profiles were used to help identify glacigenic landforms" - what hydroacoustic surveys, what additional data?

2. Unsubstantiated or unqualified interpretations.

- While the authors acknowledge that low resolution data may preclude complete landform detection, they nonetheless make interpretations of ice flow behaviour and retreat style based on the apparent absence of landforms or landform traits, especially in the southern and eastern Baltic where the input data underlying the EMODnet terrain model are sparse or entirely absent. Such interpretations are false and misleading.
  For example, in the S/E Baltic: L221, absence of cross-cutting (in fact there is an absence of lineations altogether); L273, absence of eskers; L422, absence of ploughmarks (taken as indication for land-terminating margin).
- L385: "MSGLs with locally splayed termini"... "ice streams that operated ... behind a back-stepping ice margin" - these relationships have not been demonstrated, the interpretations are unsubstantiated.
- L388: "Ice marginal signatures ... comprised overprinted lobes arising from oscillations and readvances of ice margins along with switching of flow orientations and changing lobe positions…"

This has not been demonstrated (unclear if the statement reflects the authors' own observations or relates to the reference (Kjær) provided).

- L432: "the Baltic depression is mostly floored with thick glaciolacustrine sediment" - this is not the case, demonstrated by publicly available substrate geological maps/data (e.g. EMODnet geology (Quaternary lithology) or seabed sediment classification layers; e.g. SGU 1:500,000 marine geology).
- L442-3: "the presence of lacustrine wedges and outwash fans, and large moraines in the central Baltic is consistently associated with these geological structures." This claim has not been demonstrated at all.
- L454-5: "predominance of a western ice lobe over the West Gotland Basin in the earlier stages of deglaciation and a later switch to dominance of the eastern lobe." This relative chronology has not been discussed or demonstrated.
- L455: "The suture between the two lobes was located along Gotland Island." Beyond the figure citation, this has not been demonstrated; the figure caption does not discuss this "suture".

3. Sloppy manuscript preparation

- Several instances of erroneous labelling within figures (which in some cases lead to opposing or contrasting conclusions) and mis-referencing figure numbers in the text.
  - Fig 3A: the West Gotland Basin label is incorrect - both these depressions are in the East Gotland Basin, with the topographic high Klints Bank in the middle.
  - Fig 3: "DEM conglometare margin"? Unclear what this means - data stitching boundary, data integration boundary, data seam…?
  - Fig 6: direction of flow lines in NW Skåne (green flowset) is arrowed south, instead of north (stated as north in text)
  - Fig 8: also over Skåne, the same ice-marginal line has an arrow indicating retreat both to the north and the south
  - L298: Fig 7E is offshore, not onshore as stated
  - L427: Fig 7F, I ? (Not 8)
- Unclear what the basis is for naming "phases" or "moraines" on Fig. 1B (described in the caption as "major moraine systems").
- Reference for "the Baltic Ice Lake" given as Uścinowicz 2006 - the Baltic Ice Lake has been known and named since the early 1900s, this is lazy treatment of the literature.

Sarah Greenwood, Carl Regnéll, Richard Gyllencreutz, Karol Tylmann
5 September 2023

[Figure]

**Figure:** selection of mapping errors encountered in Szuman et al. manuscript

---

## Author Comment (AC2)

We appreciate your detailed analysis of our mapping and your comments on the preprint (in black). We believe that your contribution will lead to improvement of the quality of our manuscript.

Please find our detailed response to your comments below (**in green**).

Exploitation of newly available terrain datasets in the Baltic region is undoubtedly welcome, since the offshore sector was both dynamically important to the last ice sheet and highly under-researched. However, we identify a number of concerning problems with the work presented here, notably the quality and rigour of the landform mapping. We comment only on internal issues in the manuscript. We do not discuss interpretations guided by any data not available to these authors, only what is presented as figures or text in the manuscript.

1. Mapping rigour

● We find numerous examples of erroneous landform interpretation: bedrock ridges (with little/no sediment cover), dolerite dykes, aeolian dunes, marine current bedforms, estuarine banks, among others, mapped erroneously as glacial landforms. We provide a selection of examples of mapping errors in the Figure, below, and note that these examples are just a few of those we encountered. Furthermore, with regards to mapping methods.

In general, we acknowledge some mistakes in the paper, but make the important point that mapping and interpretation in Earth Science is not an exact science, is subject to interpretations made and the time and resources available for checking and verification against other data sources (e.g. geological maps). Such sources can be very helpful and might indeed be correct in the specific cases you raise but we do not regard it universally true that previously published mapping is always the 'truth' and in many cases glacial landforms, for example, can get anchored on pre-existing bedrock structures so that they can, rather annoyingly of course, be both. Given the persistent gap in knowledge in the Baltic we were not aiming for 'definitive geological survey standard' mapping, but rather to gather enough information to act as a basis to build information about ice flow and ice margins. Indeed, we note the recent publication of Greenwood et al. (now online in press, Boreas), which reveals that although these papers proceeded entirely independent of each other, it is apparent that there is a very large degree of similarity in the findings, regarding the distribution and type of landforms identified and mapped, and especially so regarding the interpretations made of them regarding ice flow directions, sequencing and ice margins. In fact, the similarities are really rather reassuring and provide a nice example of scientific replication and with some additions and alternative interpretations that suggest both would be of value to the community.

Of specific concern are Swedish terrestrial examples where geological structures overlap with marine, lacustrine and glacial landforms. We could revise details of the mapping here, consulting data from the Swedish Geological Survey, or more simply exclude these terrestrial examples from our study.

Our mapping includes 22,500 features and reconstructions we derive are not based on single or small numbers of landforms and so even in case of misinterpretation when we do make the odd mistakes (a natural consequence of large-scale mapping – no one will be 100% correct), we still believe the conclusions hold, especially given that we concentrate on the Baltic Sea, not the terrestrial margin.

As for the seabed geology, the quality of EMODnet data also has its limits. The size of landforms is often smaller than the resolution of the geology data. In addition, geological maps indicate surficial sediments without providing detailed information on deeper compositions. There are numerous examples where post-glacial and glacial landforms occur together in the Baltic. In the example presented in Fig. F (Greenwood et al. comments to preprint), EMODnet geology does not allow for interpretation of the landform as a moraine. However, geological maps by Sviridov and Emelyanov (2000) in this region indicate moraine complexes, partly covered by sand, mud, and gravels. We interpreted their maps

and topography as indicative of an ice margin. However, considering that the geology data is inconsistent in this area we could mark the moraine as uncertain or remove.

In some locations landforms were not clearly visible on the DEM, especially where there is a mixture of high and low quality data. In our interpretations we adopted the strategy of testing each area with several hillshade orientation angles and exaggeration. We also manipulated the colour palette, adjusting to particular areas in addition to checking each non obvious landform with terrain profiles. Fig. E (Greenwood et al. comments to preprint) represent a W-E chains of higher elevated bumps. The cross-profile (Fig. 1), the arcuate shape of this chain, the close vicinity of a potential tunnel valley and an accordance with the general moraine pattern in this part of study area leads us to interpret this chain as degraded moraine.

[Figure]

*Fig. 1. Examples of cross-profiles along the chains of bumps interpreted as degraded moraine (cf. to Greenwood et al. comments to preprint Fig. E). In these cases, we see a change in elevation and texture that we interpret as degraded moraines, and which appears reasonable given the wider positioning in relation to other features and overall bathymetry. Others are free to disagree of course.*

● We find numerous examples of inconsistent mapping choices: neighbouring landforms of similar appearance mapped or unmapped.

Fig. B from Greenwood et al. comments to preprint - Agree, it is a remnant polyline that was not removed from the database after checking the mapping.

Fig. G (bottom panels) from Greenwood et al. comments to preprint - The current erosional features from Kalmar Strait (Fig. G in Greenwood et al. comments to preprint) shown as an analogue for MSGLs along Odra Bank are two different landforms. The features along Kalmar Straight, are erosional with an anastomosing pattern of grooves and irregular shape (changeable width and general geometry) of 'positive' sections. MSGLs presented by us (Fig. 7D, Szuman et al. preprint), have more or less the same width along the whole profile and a regular shape. They have a potentially erosional and depositional origin as the ridges clearly overlap each other from different azimuths.

Fig. G (upper panels) from Greenwood et al. comments to preprint –

(i) The Ronne Bank is not mapped as a wedge because of its composition (mixture of sedimentary rock, sand, till; EMODnet geological maps) and the profile shape differs from the Odra Bank. The gravel stringers were not mapped as glacial lineations as indicated in Fig. G (Greenwood et al. comments), but the positive lineations on top of the bank that correspond well with and are close to lineations in the southern part of Bornholm are. According to geological maps our lineations are anchored both in sedimentary rocks (the MSGLs occur in hard bedrock; e.g., Krabbendam et al. 2016) and soft sediments (like till, sands). Erosional glacial lineations can have very different compositions (sand, diamicton, hard rocks; e.g., Hermanowski et al. 2019). The dimensions of mapped lineations are much greater than the stringers from Fig. G in Greenwood et al. comments to preprint.

(ii) The study of Kramarska (1998) along the Odra Bank indicates the geology of the structure as of various origin including glacial. It is true that the topmost layer of the southern margin (about 5 km) of the bank comprises Littorina sands (till below). However, the study only loosely corresponds with the lineations, both topographically and by location. The lineations are present up to c. 40 km from Kramarska's (1998) data and profiles. The elevation of the lineations is 5 m lower. Please notice, that in Kramarska's (1998) C-D profile (p. 281) the top of the till layer occurs at ridges, like in our case. These ridges are not a product of sea currents activity. Most of the lineations are located on a lower terrace similar to that of the elevated terrace of the Odra Bank from Kramarska (1988) study, so it is highly probable that the topmost layer of the lower terrace comprises sediments that are buried in the upper terrace. The features have a positive topographical expression (contrary to negative examples from Kalmar in Fig. G from Greenwood et al. comments to preprint), and there are also iceberg pits present on top of the lineations. So, we prefer to keep our interpretation that these linear features are of glacial origin.

However, taking into consideration the 'waviness'/overlapping nature of the features presented in our study we can add that more detailed analyses are needed to clearly determine the origin of these landforms.

● Landforms visualised in figures have not actually been mapped (e.g. Landsort Deep, Fig. 5G, 9F).

Thanks for noting this. The landforms were mapped, but switched off for figure preparation and mistakenly omitted when preparing Fig. 4. Please notice that in Figs 5G and 9F we show this area and state that there are glacial lineations present. In Figs 5G and 9F it was our intention not to blur the figure and not to add the lineations. We would be happy to attach a corrected Fig. 4 in the revised version.

● The approach to moraine mapping offshore is very unclear: it appears that all 'bumps' not considered a lineation, rib or esker are recorded as moraines, with little/no discussion of or motivation for the interpretation. This is problematic for the discussion of the retreat pattern (e.g. Section 4.5).

We did not interpret each bump that we did not consider as lineation, rib or esker as moraine. For the interpretations we analysed topographical expression of the landform with broader context of landform assemblages. We have gathered all studies on glacial landform in the Baltic (Szuman et al. preprint, Fig.

1 and Table 1) and where possible supported our interpretation of the topographic expression using seismic surveys and hydroacoustic profiles. When combined with the variable quality of the DEM over the study area, this results in areas that are more confident and less confident in the interpretation.

● Consultation of geological maps (easily and publicly available) would have avoided many mapping errors. (Statements like L264-5 - lack of information on composition - are false.)

We agree that the mapping could be improved by consulting geological maps and, if given the chance, would do this. Given the resolution and quality of geological data and inconsistency between different published datasets (cf. EMODnet geology and Sviridov and Emelyanov 2000), interpretation of some landforms will still have some degree of uncertainty and potentially can be interpreted in different way by different people or teams. However, we also note that given the total number of landforms mapped, which we base our interpretations on, the overall amount of mistakes are insignificant and our conclusions still hold (see main point above).

● There are three versions of the tiled EMODnet DTM product (2018, 2020, 2022); the authors do not state which version was used. These versions have noticeable differences in landform visibility stemming from differences in input datasets and (re-) gridding results. Comparing with visually obvious gridding artefacts, the base topography in their Fig 4 appears to be the 2018 version - did they use more recent data too?

We based our interpretation on EMODnet 2018, as we started mapping in 2019. The newer versions, in our opinion, are not significantly different for landforms recognition. However, we would be happy to verify landforms against the 2022 product in the revised version.

● L154: "Artifacts are common in the dataset and where these occurred, cross-checking in the data from hydroacoustic surveys and seismic profiles were used to help identify glacigenic landforms" - what hydroacoustic surveys, what additional data?

We analysed different published data i.e., hydroacoustic data and seismic profiles in searching for glacial landforms. Those that were useful are mentioned in the text and/or Table 1. Additional data – e.g., cross-profiles were mentioned in the Methods section.

2. Unsubstantiated or unqualified interpretations.

● While the authors acknowledge that low resolution data may preclude complete landform detection, they nonetheless make interpretations of ice flow behaviour and retreat style based on the apparent absence of landforms or landform traits, especially in the southern and eastern Baltic where the input data underlying the EMODnet terrain model are sparse or entirely absent. Such interpretations are false and misleading. For example, in the S/E Baltic: L221, absence of cross-cutting (in fact there is an absence of lineations altogether); L273, absence of eskers; L422, absence of ploughmarks (taken as indication for land-terminating margin).

L221: In this sentence we wanted to emphasise the same direction of two groups of lineations in the east (no cross-cutting, just overprinting) and different direction of the two groups of lineations in the west (cross-cutting). We do not make a statement on the lack of lineations but rather on the replacement of cross-cutting with overlapping. Please consider sentence in L221 with the following one in L222. "In contrast, no cross-cutting relationships are identified in the SE and E part of the study area. Typically, more delicate and shorter glacial lineations overprint more prominent ones." Both sentences could be rephrased to not confuse readers.

L273: In our opinion this comment also lacks perspective based on the full paragraph on eskers (L272-282) in which we note that eskers absence could be due to poor-quality DEM data and burying processes. I.e. "We speculate that such small landforms might not be distinguishable in the poor-quality DEM

available for the offshore areas or that eskers could be buried by postglacial deposits (Uścinowicz, 1999)"

we note that

"Most of the eskers identified in this study occur onshore with only 2% of the total population located offshore."

In addition, we analyse the presence of eskers in coastal regions where good quality data are present.

We do not make interpretations of ice flow behaviour and retreat style based on the apparent absence of landforms. By including statements on the lack of landforms we instead look to emphasise that interpretations in the eastern Baltic are not strong and in need of better-quality data.

L422: In this sentence we indicate that ploughmarks are missing in the southern sector of the study region but are present in the north, and that this likely reflects a transition from a land-terminating to shallow lacustrine (aqueous) calving margin. It is true that here we make an interpretation based on a transition between absence and presence of landforms. However, since our southern sector comprises areas with high quality data where we are confident in our interpretations, we do not see what is false in this statement.

● L385: "MSGLs with locally splayed termini"… "ice streams that operated … behind a back-stepping ice margin" - these relationships have not been demonstrated, the interpretations are unsubstantiated.

In line 384 we refer reader to Figs 6 and 9C where locally splaying termini are present. In particular, we refer readers to inspect flowsets presented in Fig. 6 and our lineation mapping (Fig. 4 and partly in Fig. 6). We therefore do think that we identify locally splaying termini. We will refer more clearly to Fig. 4 in L384 to help clarify the statement.

● L388: "Ice marginal signatures … comprised overprinted lobes arising from oscillations and readvances of ice margins along with switching of flow orientations and changing lobe positions…" This has not been demonstrated (unclear if the statement reflects the authors' own observations or relates to the reference (Kjær) provided).

We do not understand this comment as there is no cited sentence present in L388. Possibly this refers to L398.

In our opinion, the study of Kjaer et al. (2003) is consistent with our manuscript. We could not provide landform level detail for every statement in the discussion. However, we provide the reader (in addition to numerous landform examples) with the mapping in Fig. 4, flowsets in Fig. 6 and possible margin retreat scenario in Fig 8. In particular, the flowsets (Fig 6) and landforms (Fig. 4) demonstrate flow switching and overprinting in the southern sector (zoom in Fig. 9B). We could clarify this by adding reference to those figures and to other publications with consistent statements (e.g. Gehrmann & Hardig 2018; Pedersen 2000).

We suggest replacing the sentence with a new one to better clarify what we mean: "Ice marginal signatures in the southern sector often comprise overprinted lobes arising from ice-margin oscillations and readvances (Figs 6, 9B) along with switches in flow orientation and changing lobe positions during overall retreat (cf. Pedersen 2000; Kjær et al., 2003; Gehrmann & Hardig 2018)"

● L432: "the Baltic depression is mostly floored with thick glaciolacustrine sediment" - this is not the case, demonstrated by publicly available substrate geological maps/data (e.g. EMODnet geology (Quaternary lithology) or seabed sediment classification layers; e.g. SGU 1:500,000 marine geology).

We agree, that 'thick' is a relative statement, as some areas have 5-15 meters of unconsolidated sediments (see e.g., Flodén 1997; Tulling and Flodén 2001; Sopher 2016) the others more than 20 m

(e.g. Bjork 1990; Lemke 1995, 1998; Sopher 2016; Kramarska 2016). We will change this sentence to better correspond to these data presented in the geological datasets.

● L442-3: "the presence of lacustrine wedges and outwash fans, and large moraines in the central Baltic is consistently associated with these geological structures." This claim has not been demonstrated at all.

We agree that this statement could be better related to presented data. We do not provide a direct figure showing this, however, the statement is implicit when inspecting Figs 1, 4, 7E, 9D. We will provide more description in this paragraph in order to clarify.

● L454-5: "predominance of a western ice lobe over the West Gotland Basin in the earlier stages of deglaciation and a later switch to dominance of the eastern lobe." This relative chronology has not been discussed or demonstrated.

We provide relative timing based on superimposition of the landforms in Fig. 6C based on flowsets at Gotland where the density of the landforms is high, providing strong evidence for the statement. We indicate in the results that cross-cutting in Gotland is common (L219-220). We comment on it in the flowset section (L252-253).

"Overprinting (e.g., Figs 6B, C), on Gotland Island records 10 flowsets with orientations switching from N-S to NNE-SSW oriented, through NE-SW, ENE-WSW, and NW-SE (Fig. 6C)"

● L455: "The suture between the two lobes was located along Gotland Island." Beyond the figure citation, this has not been demonstrated; the figure caption does not discuss this "suture".

The reference here should be to Figs 8; 9C, E, F. Description of the suture zone could be added or rephrased from interstream zone, and better clarified on Fig. 9E.

3. Sloppy manuscript preparation

● Several instances of erroneous labelling within figures (which in some cases lead to opposing or contrasting conclusions) and mis-referencing figure numbers in the text.

○ Fig 3A: the West Gotland Basin label is incorrect - both these depressions are in the East Gotland Basin, with the topographic high Klints Bank in the middle.

Only East Gotland Basin should be included here.

○ Fig 3: "DEM conglometare margin"? Unclear what this means - data stitching boundary, data integration boundary, data seam…?

This label could be changed to data stitching boundary in order to clarify.

○ Fig 6: direction of flow lines in NW Skåne (green flowset) is arrowed south, instead of north (stated as north in text)

Direction of flow lines in NW Skåne (green flowset) is arrowed south.

○ Fig 8: also over Skåne, the same ice-marginal line has an arrow indicating retreat both to the north and the south

Here, there are 2 lines, the first indicates recession toward the SW, the second toward the N-NE, but it is true that they overlap each other. We will include a bigger space between the lines.

○ L298: Fig 7E is offshore, not onshore as stated

Thanks. Reference to figure should be offshore.

○ L427: Fig 7F, I ? (Not 8)

Should be Fig.7

● Unclear what the basis is for naming "phases" or "moraines" on Fig. 1B (described in the caption as "major moraine systems").

Basis is for naming "phases" or "moraines" – should be unified to phases

● Reference for "the Baltic Ice Lake" given as Uścinowicz 2006 - the Baltic Ice Lake has been known and named since the early 1900s, this is lazy treatment of the literature. Sarah Greenwood, Carl

Reference for "the Baltic Ice Lake" – the amount of references could potentially be extended, but it is not necessary to cite the paper from 1900s.

In summary, we think that the most of the Greenwood et al. comments to preprint are technical and editorial rather than substantial for our general reconstruction and interpretations. Indeed the Greenwood et al. (now online in press, Boreas) paper published in Boreas independently has very similar findings to our results.

Kind regards,

Izabela Szuman-Kalita,
Jakub Kalita
Christiaan Diemont
Stephen Livingstone
Chris Clark
Martin Margold

---

## Author Comment (AC3)

Dear Reviewer,

We appreciate your detailed comments on the preprint (in black). We believe that your contribution will lead to improvement of the quality of our manuscript. Please find our detailed response to your comments below (in green).

General comments:

This study is an ambitious and laborious task mostly based on combination of bathymetric data sets of different resolution supported by data from earlier offshore studies and "onshore literature". Approaches to utilize increasing amount of high-resolution bathymetric data are crucial to promote scientific discussion on the distribution of submarine glaciogenic features and the influence of sea-bed bedrock/topography on glacial dynamics, especially in the case of streaming ice. The manuscript will add to progress in numerical modelling of the Scandinavian or Fennoscandian ice sheet.

In general, structure and proper length of the manuscript make it easy to crasp. However, the major problem with the manuscript is the quality of the mapping procedure with some likely misinterpretations or lack of alternative explanations in the presented data, as also indicated by Greenwood et al. in their interactive comments. Moreover, like commented by Greenwood et al., the use of geological maps for coastal periphery-offshore interpretations would have increased the quality of the methodological approach. Uncertain observations on maps should be better presented/visualized and reasoning/data behind some interpretations, regarding f. ex. moraine landforms, should be better explained. It is important that the results and "dynamic history of the Baltic sector of the SIS" presented in Figure 8 are based on "solid" ground. Nevertheless, it is likely that some mistakes or misinterpretations remain in these kinds of maps based on large data sets with varying resolution. To sum up, the results presented in the manuscript do not convincingly enough support the interpretations made. I think the manuscript should be published after major revision, answering to above-mentioned main concerns (as also interactively commented by Greenwood et al.), and including reevaluation of some key observations or interpretations pointed out in the specific comments.

The idea of mapping procedure is to map as many features as possible to build a larger story – from single landform to flowset, to single ice stream, to particular margin, to retreat pattern, etc. We agree, that in some cases our final database of mapped features still contains some mistakes. But the biggest advantage of the mapping approach is that even if we lose one piece (misinterpreted landform) the other pieces show us the final puzzle image/story. Our mapping includes 22,500 features, and so even if there are a few mistakes (a natural consequence of large-scale mapping – no one will be 100% correct), we still believe the conclusions hold. We were not aiming for 'definitive geological survey standard' mapping, but rather to gather enough information to act as a basis to build information about ice flow and ice margins. Indeed, we note the recent publication of Greenwood et al. (now online in press, Boreas) - although these papers proceeded entirely independent of each other, it is apparent that there is a very large degree of similarity in the findings. Thus, we believe the landform misinterpretations are more technical problems that do not substantially affect the main conclusions of our paper.

We provide some arguments for our interpretations in replies to Greenwood et al. comments to preprint, but we agree that we can add alternative explanations where interpretations are less certain. In general, our conclusions are consistent with Greenwood et al. (now online in press, Boreas) and we acknowledge some misinterpretations e.g., diorite dyke and esker, we could fix in a revised version. The important conclusions we found are e.g.: (i) evidence for narrower corridors of fast ice flow that we interpret as distinct and smaller ice streams instead of 300-km wide zone of accelerated ice; (ii) episodic rather than steady ice retreat; (iii) our mapping identifies flow and ice marginal geometries from both Swedish and north Bothnian sources. These are not the issues obtained from analysis of single landform, and their detail geology, but a regional scale overview of landform distribution and their aggregation into flowsets, margins and patterns.

Please see also replies to Greenwood et al. comments to preprint.

Specific comments

Title: the title is ok but not very "selling" in terms of the glacial dynamics or methodology used.

Our proposition is to change the title from "Evaluation of the role of the Baltic depression during deglaciation of the last Scandinavian Ice Sheet; a landform-driven investigation" to "Evaluation of the role of the Baltic depression in controlling the dynamics of the last Scandinavian Ice Sheet; a seabed and coastal landform-driven investigation"

L_16 (Abstract): … "that might have resulted in rapid collapse of this ice sheet sector." <> possible rapid collapse of the ice sheet is not discussed in the main text – it should be removed from the abstract or discussed!

Agree with that. Will be removed

L_37: I would not use the term conduit here, very broad ´channel´ is simple enough.

Agree.

L_38: it would be relevant to include information/references on the shifting ice-divide along the Scandes mountains and its probable influence on the ice dynamics!

Agree. Our proposition is to add reference to Boulton et al., 1985; Kleman et al., 1997; Patton et al. 2016.

L_39: you should maybe use more accurate definition here (unconsolidated). There is also different behavior/effect of soft sediments, depending on their composition and grainsize distribution.

Agree. Unconsolidated is better.

L_41-42: "impeding water evacuation from the ice margin" <> not from the ice-margin but from the proglacial basin?

Agree.

L_53: I think this research question is not really relevant here, the Baltic depression surely had an overall, prominent role for LGM position but related details are more poorly known + in the end you also answer first to the second question which apparently is the main issue.

Agree. Could be removed.

Table 1: Feldens et al., Dorokhov et al., Schäfer et al., Jensen, All et al. and Jakobsson et al. 2016, 2020 are not mentioned/cited in the text. If these studies are important for the landform interpretations, they should be mentioned!

Agree. Appropriate references will be added.

Figure 2: "No submarine evidence was available at the time Boulton et al. (2001) drew the figure and it is a purpose of this paper to seek information about the footprint of this ice stream and its likely width." <> this sentence is simply well put and would be good to replace it into the introduction as well?

Agree. Will be added to introduction.

L_125-130: "Our motivation for mapping the coastal periphery of the Baltic is twofold:

1. It eases integration with the much-studied onshore landform record,

2. To provide a high-resolution verification for ice-flow patterns and ice-marginal retreat patterns mapped from the lower quality and resolution bathymetric dataset.

<> would it be better to place this text into Methods?

Agree.

L_142: zones of soft sediment (similar consolidation or properties?)

Unconsolidated is better.

L_144: could these fan-shaped belts be indicated on a map?

Agree. Probably best place to add them would be Fig. 1

L_146: soft sediments has been found ... <> does this mean that elsewhere in the basin there are no soft sediments?

"The accumulation of soft sediments …" could be changed to "The accumulation of soft sediments has been found to especially correlate…"

L_151: cross-checking <> this is a bit unclear way to express this – where there always hydroacoustic or seismic data available for artifact recognition? – somehow related to works presented in Table 1?

Reference to Table 1 should be added here. Cross-checking was possible when hydroacoustic and seismic data was available, not for every artifact. Sentence should be rephrased.

L_157: Mapping was carried out ... <> in this case the use of geological maps to verify interpretations (where possible) would have been necessary. It is important to include information from geological maps to methods as it likely strengthens the interpretations along the coastline-offshore regions. Also Greenwood et al. raise this point in their interactive comments!

We agree with that. Please see our reply to the Greenwodd et al. comments to preprint.

Table 2: for the future work, in case there are changes to databases, it would be good to indicate the date when the data was collected (f.ex. 02/2023)

Agree.

L_168: you mean glaciolacustrine wedges, please explain the term shortly when mentioned for the first time.

Yes. Glaciolacustrine. "(ice-lake contact depositional wedge-like fan)" could be added.

L_170: tunnel valleys <> do these include possible (braided) subglacial meltwater routes or corridors not strictly classified as tunnel valleys? See possible route in Fig. 3B just below the text: good quality DEM?

No. Our tunnel valley mapping does not include these features.

L_179-180: "identifying larger features in these regions including moraines, ice marginal channels, and ice marginal deposits was often still possible" <> what are these interpretations based on? – any references/supporting earlier data or just interpretation based on ...?

This sentence will be rephrased to include information that interpretation was possible for landforms where the size of the landform is greater than the resolution of the DEM and is visible in these data (e.g. due to hillshade angle, z-factor, color palette), profiles or other studies indicating the landform are present.

Figure 3: 3C) offshore area west of the Gdansk Bay has been mapped to show marginal meltwater channels (Fig 4) <> why not longshore structures (bars & troughs) related to a large spit-platform?

The profiles correspond more to marginal meltwater channels than to longshore structures. The scale of longshores structures vs. resolution of available DEM make the recognition of longshore structures difficult. Marginal meltwater channel interpretation is in accordance with meltwater routes indicated by Uścinowicz 1999 with seismic profile not directly on those landforms but present nearby. But, the smaller features could be changed to speculative meltwater channels.

L_200: remove likely

Agree.

Figure 4: would it be possible to explain in the text why there are so little landforms within the offshore areas located SW and NE from the Landsort Deep?

We agree that reader would benefit from such information. Our proposition is to add such information to the figure caption. Small numbers of landforms coincide with the presence of hard bedrock.

L_232: Fig. 5B <> where are ribbed moraines in this figure? - wrong fig?

Ribbed moraines are located in NW part of the figure. We should add this reference inside the figure.

Figure 5G: these landforms are not presented on a map!

Agree. We should correct this.

L_261: (Fig. 7E) <> I think that this ice-marginal moraine interpretation is not well justified - any reference from earlier marine investigations or reasoning behind the interpretation?

This area is poorly investigated. However, one of the moraines at Southern Middle Bank is present in the profile by Uścinowicz 1999 (Fig 5). The distal part of the moraine is interpreted as a glaciofluvial delta. We should add this reference to the sentence in L261, as follow:

Prominent end moraines are found throughout the study region including the central Baltic where they are up to 30 m high and 45 km wide (Fig. 7E). Uścinowicz (1999) interpreted the central feature from Fig. 7E as glaciofluvial delta.

L_267: (Fig. 7D) <> in this figure?

Should be 7E. Label with Słupsk Bank should be added to the figure.

L_268: distinct (?) + inferred (Fig. 7E), with <> I can´t see this in Fig. 7E

Flow directions are indicated in Fig. 7E by black arrows.

Figure 6B: the leftmost flowset probably has a wrong arrow direction

No, it is correct. The lineations indicate ice flow from the north toward the south.

L_301: geological structures <> can you be a bit more precise?

We think geological structures is fine.

L_308: "Iceberg pits are identified in one locality" <> any reference for the interpretation, what is this interpretation based on?

The landform interpreted as iceberg pits are about 100-200 m long with amplitudes c. 1.5 and 2 m. Often the distal side is higher than the proximal side (with regard to inferred ice flow direction; see Fig

7D). Potentially they can be interpreted as push mounds. Such an interpretation is also consistent with the glacial context of the area (see next comment).

Figure 7D: I´m not convinced that these are glacial lineations! - too non-uniform + winding/overlapping - for me these landforms rather resemble submarine erosional/depositional features by bottom currents!

We agree, that the presented linear features are a combination of erosion and deposition (some above some below the bottom surface). They also overlap each other from three directions, more or less NE-SW. The study of Kramarska (1998) indicates the geology of the structure as of various origin including glacial. It is true that the topmost layer of the southern margin (about 5 km) of the bank comprises Littorina sands (till below). However, the study only loosely corresponds with the lineations, both topographically and by location. The lineations are present up to c. 40 km from Kramarska's (1998) data and profiles. The elevation of the lineations is 5 m lower. Please notice, that in Kramarska's (1998) C-D profile (p. 281) the top of the till layer occurs at ridges, like in our case. These ridges are not a product of sea currents activity. Most of the lineations are located on a lower terrace similar to that of the elevated terrace of the Odra Bank from Kramarska (1988) study, so it is highly probable that the topmost layer of the lower terrace comprises sediments that are buried in the upper terrace. The features have a positive topographical expression (contrary to negative examples from Kalmar in Fig. G from Greenwood et al. comments to preprint), and there are also iceberg pits present on top of the lineations. So, we prefer to keep our interpretation that these linear features are of glacial origin.

However, taking into consideration the 'waviness'/overlapping nature of the features presented in our study we can add that more detailed analyses are needed to clearly determine the origin of these landforms.

Figure 7F: why are these interpreted as moraines, any earlier supporting data?

An ice margin is interpreted to be at this location according to the study of Noormets, & Flodén (2002b) and their seismic profiles. The DEM is of good quality in this location. Notice in Fig. 4 that there are lots of moraines forming a group in this part of Baltic. There are MSGLs not far away at the proximal side. Additionally, the profiles are consistent with moraine profiles:

[Figure]

[Figure]

Figure 8: although there likely will remain some erroneous interpretations/details of landforms in this massive data set, this figure (black lines) should stand strong on hard evidence or if black lines are based on uncertain or extrapolated data, they should be marked with f.ex. dashed lines

In our opinion the landforms classification into two groups i.e., certain and uncertain landforms gives sufficient insight into uncertainty. Adding additional gradation of uncertainty to landforms derivatives would not help to build more advanced interpretations.

L_403: 9B <> not shown in the figure

"glaciolacustrine wedges" reference should be added to Fig. 9B

L_405-406: "An example of this is in Pomeranian Bay where a glaciolacustrine fan is overprinted by long cross-cutting MSGLs, indicating a subaquatic origin." <> sorry, but I do not understand this sentence (or reasoning), and how does it relate to previous sentence. MSGLs refer to ice flow over the wedges BUT I think these MSGLs are not glacial features at all but instead submarine current structures!

Sentence should be changed to "An example of this is in Pomeranian Bay."

L_416: prominent moraines <> based on what? - any other supporting data than DEM interpretation?

Part of prominent moraines are supported by study of Uścinowicz (1999).

L_427: (Figs 8F, I; <> wrong figure reference

Agree. Should be Figs. 7F, I

L_436: outwash fans <> these not mapped in the present study?

The outwash fans from this sentence refers to the study of Noormets and Flodén (2002b) not to our mapping that do not identify outwash fans.

L_437: moraine interpretation based on what?

This interpretation is based on the DEM (expression of the landform and its context) and supported by Noormets and Flodén, (2002b)

L_450-452: "We confirm this Bothnian ice source from the north, but also identify a Swedish ice source with a ~NW-SE orientation based on our ice marginal and flowset geometries (Figs 6, 8, 9C)." <> just to check this - are you absolutely sure that you are the first ones to report this Swedish ice source in this area?

We are pretty certain, although we checked only English written literature. In fact Noormets and Flodén (2002b) state that area north of Gotska Sandon was an inter-stream area separating ice lobe located in the Landsort Deep from that in the Faro Deep. They however do not mention that each had a separate source. A Swedish ice source can be implicitly deducted from their study.

L_457-458: Bedforms within the Landsort Deep are not presented on map in Fig 4! The Deep is an interesting and outstanding submarine geological feature, so it would be good to discuss its origin (+ references) - could there be any alternative explanation for the features interpreted here as glacial lineations?

Agree. We should comment more on this point. Our opinion is that this interpretation has a strong basis. Considering the quality of DEM (Fig 5G), context (associated moraines and nearby inter-stream supported by Noormets and Flodén (2002b) ), sediments (partly coinciding with glacial clays) and geology (pre-quaternary sandstones contrary to surrounding hard bedrock supported by All et al. (2006)) these can be interpreted as glacial features. Greenwood et al. (in press now, Boreas), independently, also interpreted these landforms as glacial lineations.

Technical corrections

We thank the reviewer for going through the manuscript in detail and indicating technical corrections which we think should be included in text.

---

## Author Comment (AC4)

Dear Reviewer,

We appreciate your comments on the preprint (in black). We believe that your contribution will lead to improvement of the quality of our manuscript. Please find our detailed response to your comments below (**in green**).

General comments:
This study is ambitious and the specific topic is highly relevant for the glacial dynamical development of the Fennoscandian ice sheet. The manuscript is of a suitable length and its structure is proper, making it easy to crasp. However, I agree with the comments of the anonymous referee and interactive comments by Greenwood et al. The mapping procedure quality issues (misinterpretations), lack of use of geological maps and the need for re-evaluation of some interpretations cause a need for major revisions.

The idea of mapping procedure is to map as many features as possible to build a larger story – from single landform to flowset, to single ice stream, to particular margin, to retreat pattern, etc. We agree, that in some cases our final database of mapped features still contains some mistakes. But the biggest advantage of the mapping approach is that even if we lose one piece (misinterpreted landform) the other pieces show us the final puzzle image/story. Our mapping includes 22,500 features, and so even if there are a few mistakes (a natural consequence of large-scale mapping – no one will be 100% correct), we still believe the conclusions hold. We were not aiming for 'definitive geological survey standard' mapping, but rather to gather enough information to act as a basis to build information about ice flow and ice margins. Indeed, we note the recent publication of Greenwood et al. (now online in press, Boreas) - although these papers proceeded entirely independent of each other, it is apparent that there is a very large degree of similarity in the findings. Thus, we believe the landform misinterpretations are more technical problems that do not substantially affect the main conclusions of our paper.

Specific comments:
(in order to minimize duplication, I will mainly comment only those points not mentioned by Greenwood et al. or Anonymous referee).

L25 (Abstract): "…broad changes in ice flow geometry, ranging from SE-NW to N-S and then to NWSE."
<> Ice flow direction replacing ice flow geometry? And maybe with description of the flow areas, for example: SE-NW in the western (SW of Malmö) area.

Thank you for that comment, we would correct this.

L139_L146: terminology to be clarified: The unconsolidated sediments versus sedimentary rocks. Also it is unclear what is meant with "soft sediments"

Agree, this should be changed to unconsolidated when writing about sand and till, and sedimentary rocks when relevant.

L150: Elevation data => Bathymetric data?

Yes, would be changed.

L205: Soft sediment => unconsolidated sediment?

Here, soft sedimentary rocks and unconsolidated sediments. We would change this to make the sentence more specific.

Technical corrections:

L77: "…from the south…"=> from towards the south
L134 and L138: "…north-western Baltic…" and "…SE Baltic…" => north-western Baltic main basin and
SE Baltic main basin?

Figure 3B and 3D: conglometare => conglomerate (Is there a better word, for example bathymetric data type margin etc.?)

L429: could passed => could have passed

Thank for all the detailed technical correction, that will be implemented in the revised version.

Best regards,

Izabela Szuman-Kalita,
Jakub Kalita
Christiaan Diemont
Stephen Livingstone
Chris Clark
Martin Margold

---

## Author Response (AR1)

Dear Reviewer 1,

We appreciate your detailed comments on the preprint (in black). We believe that your contribution will lead to improvement of the quality of our manuscript. Please find our detailed response to your comments below (in green).

General comments:

This study is an ambitious and laborious task mostly based on combination of bathymetric data sets of different resolution supported by data from earlier offshore studies and "onshore literature". Approaches to utilize increasing amount of high-resolution bathymetric data are crucial to promote scientific discussion on the distribution of submarine glaciogenic features and the influence of sea-bed bedrock/topography on glacial dynamics, especially in the case of streaming ice. The manuscript will add to progress in numerical modelling of the Scandinavian or Fennoscandian ice sheet.

In general, structure and proper length of the manuscript make it easy to crasp. However, the major problem with the manuscript is the quality of the mapping procedure with some likely misinterpretations or lack of alternative explanations in the presented data, as also indicated by Greenwood et al. in their interactive comments. Moreover, like commented by Greenwood et al., the use of geological maps for coastal periphery-offshore interpretations would have increased the quality of the methodological approach. Uncertain observations on maps should be better presented/visualized and reasoning/data behind some interpretations, regarding f. ex. moraine landforms, should be better explained. It is important that the results and "dynamic history of the Baltic sector of the SIS" presented in Figure 8 are based on "solid" ground. Nevertheless, it is likely that some mistakes or misinterpretations remain in these kinds of maps based on large data sets with varying resolution. To sum up, the results presented in the manuscript do not convincingly enough support the interpretations made. I think the manuscript should be published after major revision, answering to above-mentioned main concerns (as also interactively commented by Greenwood et al.), and including reevaluation of some key observations or interpretations pointed out in the specific comments.

The improved version of our manuscript has some changes in landform mapping, slight changes in margin and flowset positions according to geological maps we consulted and a newer version of EMODnet 2022 implementation in mapping process. However our approach was focused on geomorphological expression of landforms not on their surface geology e.g., if the ridge met the requirements of moraine ridge (fan shaped, perpendicular to the ice flow direction and, in ideal situation, associated to other landforms) even if it was covered by e.g., marine sand it was classified as a moraine. We were not aiming for 'definitive geological survey standard' mapping, but rather to gather enough information to act as a basis to build information about ice flow and ice margins. Indeed, we have now seen the recent publication of Greenwood et al. (now online in press, Boreas) - although these papers proceeded entirely independent of each other, it is apparent that there is a very large degree of similarity in the distribution, pattern and classification of landforms. Thus, we regard the stated landform misinterpretations were more technical problems and that do not substantially affect the main conclusions of our paper, and which we retain.

We included also more alternative interpretations and refereed to the earlier studies on glacial landforms along Baltic Basin. We added also more details into the *method section* to explain some details of our approach. In our revised version we replaced *moraines* by 'ice-contact landforms', that include moraines, ice-contact fans and grounding zone wedges, and expanded their interpretations referring to some examples. Palaeoglaciologically it means an ice margin lay here without us having to intrepet the exact process that formed the landforms. This resolves the classification problem raised by referees.

Specific comments

Title: the title is ok but not very "selling" in terms of the glacial dynamics or methodology used.

The final proposition is: *Reconstructing dynamics of the Baltic Ice Stream Complex during deglaciation of the Last Scandinavian Ice Sheet*.

L_16 (Abstract): … "that might have resulted in rapid collapse of this ice sheet sector." <> possible rapid collapse of the ice sheet is not discussed in the main text – it should be removed from the abstract or discussed!

Removed.

L_37: I would not use the term conduit here, very broad ´channel´ is simple enough.

Agree, changed.

L_38: it would be relevant to include information/references on the shifting ice-divide along the Scandes mountains and its probable influence on the ice dynamics!

We add reference to Boulton et al., 1985; Kleman et al., 1997; Patton et al. 2016:

Ice dynamics and changes in ice flow direction were also linked to ice divide shifting from the Scandinavian mountains to over the Bothnian Basin (Boulton et al., 1985; Kleman et al., 1997; Patton et al., 2016).

L_39: you should maybe use more accurate definition here (unconsolidated). There is also different behavior/effect of soft sediments, depending on their composition and grainsize distribution.

Changed to unconsolidated.

L_41-42: "impeding water evacuation from the ice margin" <> not from the ice-margin but from the proglacial basin?

Agree, changed.

L_53: I think this research question is not really relevant here, the Baltic depression surely had an overall, prominent role for LGM position but related details are more poorly known + in the end you also answer first to the second question which apparently is the main issue.

Agree, removed.

Table 1: Feldens et al., Dorokhov et al., Schäfer et al., Jensen, All et al. and Jakobsson et al. 2016, 2020 are not mentioned/cited in the text. If these studies are important for the landform interpretations, they should be mentioned!

Agree, added.

Figure 2: "No submarine evidence was available at the time Boulton et al. (2001) drew the figure and it is a purpose of this paper to seek information about the footprint of this ice stream and its likely width." <> this sentence is simply well put and would be good to replace it into the introduction as well?

Agree, added to introduction.

L_125-130: "Our motivation for mapping the coastal periphery of the Baltic is twofold:

1. It eases integration with the much-studied onshore landform record,

2. To provide a high-resolution verification for ice-flow patterns and ice-marginal retreat patterns mapped from the lower quality and resolution bathymetric dataset.

<> would it be better to place this text into Methods?

*Agree, moved to methods.*

L_142: zones of soft sediment (similar consolidation or properties?)

*Changed to unconsolidated.*

L_144: could these fan-shaped belts be indicated on a map?

*They are added as a result of our mapping in Fig. 4 and following Figs, however indicated by lines to minimalise uncertainty effect due to DEM resolution.*

L_146: soft sediments has been found ... <> does this mean that elsewhere in the basin there are no soft sediments?

*Changed to: Accumulations of unconsolidated glacifluvial and glacial sediments particularly correlate with the position of sills and banks (Kramarska, 1998; Lemke and Kuijpers, 1995; Obst et al., 2017; Uścinowicz, 1999; Noormets and Flodén, 2002a).*

L_151: cross-checking <> this is a bit unclear way to express this – where there always hydroacoustic or seismic data available for artifact recognition? – somehow related to works presented in Table 1?

*Changed to: Artifacts are common in the dataset (see e.g., Fig. 3B) and its cross-checking was possible when hydroacoustic and seismic data (Table 1) were available.*

L_157: Mapping was carried out ... <> in this case the use of geological maps to verify interpretations (where possible) would have been necessary. It is important to include information from geological maps to methods as it likely strengthens the interpretations along the coastline-offshore regions. Also Greenwood et al. raise this point in their interactive comments!

*Done.*

Table 2: for the future work, in case there are changes to databases, it would be good to indicate the date when the data was collected (f.ex. 02/2023)

*Agree, done.*

L_168: you mean glaciolacustrine wedges, please explain the term shortly when mentioned for the first time.

*Changed to broader term ice-contact fan.*

L_170: tunnel valleys <> do these include possible (braided) subglacial meltwater routes or corridors not strictly classified as tunnel valleys? See possible route in Fig. 3B just below the text: good quality DEM?

*Our tunnel valley mapping does not include these features.*

L_179-180: "identifying larger features in these regions including moraines, ice marginal channels, and ice marginal deposits was often still possible" <> what are these interpretations based on? – any references/supporting earlier data or just interpretation based on ...?

Changed to: However, it was often possible to detect larger features in these regions when the size of the landform was greater than the resolution of the DEM and by investigating multiple variants of hill-shade angles, z-factors, colour palettes, terrain profiles and comparing them with available studies (Table 1).

Figure 3: 3C) offshore area west of the Gdansk Bay has been mapped to show marginal meltwater channels (Fig 4) <> why not longshore structures (bars & troughs) related to a large spit-platform?

The scale of longshore structures vs. resolution of available DEM make the recognition of longshore structures difficult. Marginal meltwater channel interpretation is in accordance with meltwater routes indicated by Uścinowicz 1999 with seismic profile not directly on those landforms but present nearby. We changed smaller features to speculative meltwater channels.

L_200: remove likely

Done.

Figure 4: would it be possible to explain in the text why there are so little landforms within the offshore areas located SW and NE from the Landsort Deep?

Added to the text: The densest concentration of glacial lineations were mapped in Landsort Deep, where they are oriented N-S and NW-SE, with the latter being more dominant. They occupy pre-quaternary sandstones, contrary to surrounding hard bedrock to the north, east and west (All et al. 2006).

L_232: Fig. 5B <> where are ribbed moraines in this figure? - wrong fig?

Indicated in Fig.

Figure 5G: these landforms are not presented on a map!

Added.

L_261: (Fig. 7E) <> I think that this ice-marginal moraine interpretation is not well justified - any reference from earlier marine investigations or reasoning behind the interpretation?

Changed to: The most prominent ice-contact landforms are found in the central Baltic where they are up to 30 m high and 45 km long (Fig. 7E), with steep proximal and gentle distal slope. The only information on their sedimentological composition comes from limited seismo-acoustic profiles, which indicate that landforms are composed of till and glaciofluvial sediments (Uścinowicz 1999). The cross-profiles combined with the shallow water in the region during deglaciation (Fig. 7E) indicate these are not grounding zone wedges but ice-contact fans.

L_267: (Fig. 7D) <> in this figure?

Changed to 7E and a name Słupsk Bank added to Fig.

L_268: distinct (?) + inferred (Fig. 7E), with <> I can´t see this in Fig. 7E

Flow directions are indicated in Fig. 7E by black arrows.

Figure 6B: the leftmost flowset probably has a wrong arrow direction

Thank you! Yes, changed.

L_301: geological structures <> can you be a bit more precise?

We think geological structures is fine.

L_308: "Iceberg pits are identified in one locality" <> any reference for the interpretation, what is this interpretation based on?

Removed from our mapping.

Figure 7D: I´m not convinced that these are glacial lineations! - too non-uniform + winding/overlapping - for me these landforms rather resemble submarine erosional/depositional features by bottom currents!

We added more text and classified them as uncertain:

The lineations are interpreted to be glacial in origin, but with low confidence given their waviness and overlapping nature. They could also be formed by bottom currents or a mixture of glacial and non-glacial processes (i.e., exposed by erosional activity of bottom currents). Klingberg and Larsson (2017) identified lineations eroded by currents west of Öland Island, however they are clear erosional landforms with an anastomosing pattern of grooves and irregular shape (changeable width and general geometry) of 'positive' sections. MSGLs in Pomeranian Bay (Fig. 7D) have more or less the same width along the whole profile and a regular shape. They have a potentially erosional and depositional origin as the ridges clearly overlap each other from different azimuths, and they represent positive and negative landforms. In the vicinity (point 1 in Fig. 1B) there are ridges composed of glacial till with lineation-like cross profiles (Kramarska, 1998), buried in postglacial sands. Further detailed studies are required to confidently interpret the landforms in Pomeranian Bay.

Figure 7F: why are these interpreted as moraines, any earlier supporting data?

An ice margin is interpreted to be at this location according to the study of Noormets, & Flodén (2002b) and their seismic profiles. The DEM is of good quality in this location. There are MSGLs not far away at the proximal side. The landform the most to the left has typical appearance for moraine. The others could be as grounding zone wedges, with typical gentle proximal and steep distal slope.

Figure 8: although there likely will remain some erroneous interpretations/details of landforms in this massive data set, this figure (black lines) should stand strong on hard evidence or if black lines are based on uncertain or extrapolated data, they should be marked with f.ex. dashed lines

In our opinion the landforms classification into two groups i.e., certain (black) and uncertain (gray) landforms gives sufficient insight into uncertainty. Adding additional gradation of uncertainty to landforms derivatives would not really help in our opinion.

L_403: 9B <> not shown in the figure

Reference added to Figs 7D, 9B.

L_405-406: "An example of this is in Pomeranian Bay where a glaciolacustrine fan is overprinted by long cross-cutting MSGLs, indicating a subaquatic origin." <> sorry, but I do not understand this sentence (or reasoning), and how does it relate to previous sentence. MSGLs refer to ice flow over the wedges BUT I think these MSGLs are not glacial features at all but instead submarine current structures!

Changed. We also added alternative explanation and are now more speculative regarding their origin as you suggest so we present them on Fig. 4 as uncertain.

The lineations are interpreted to be glacial in origin, but with low confidence given their waviness and overlapping nature. They could also be formed by bottom currents or a mixture of glacial and non-glacial processes (i.e., exposed by erosional activity of bottom currents). Klingberg and Larsson (2017) identified lineations eroded by currents west of Öland Island, however they are clear erosional landforms with an anastomosing pattern of grooves and irregular shape (changeable width and general geometry) of 'positive' sections. MSGLs in Pomeranian Bay (Fig. 7D) have more or less the same width along the whole profile and a regular shape. They have a potentially erosional and depositional origin as the ridges clearly overlap each other from different azimuths, and they represent positive and negative landforms. In the vicinity (point 1 in Fig. 1B) there are ridges composed of glacial till with lineation-like cross

profiles (Kramarska, 1998), buried in postglacial sands. Further detailed studies are required to confidently interpret the landforms in Pomeranian Bay.

L_416: prominent moraines <> based on what? - any other supporting data than DEM interpretation?

Changed to ice-contact fans based on steeper proximal and more gentle distal slope, the size and lithology provided by Uścinowicz (1999).

L_427: (Figs 8F, I; <> wrong figure reference

Thanks ! Agree, changed to Fig. 7F.

L_436: outwash fans <> these not mapped in the present study?

The outwash fans from this sentence refers to the study of Noormets and Flodén (2002b).

L_437: moraine interpretation based on what?

This interpretation (the westernmost) is based on the DEM (expression of the landform and its context) and their unconsolidated nature is supported by Noormets and Flodén, (2002b). The other are marked as grounding zone wedges.

L_450-452: "We confirm this Bothnian ice source from the north, but also identify a Swedish ice source with a ~NW-SE orientation based on our ice marginal and flowset geometries (Figs 6, 8, 9C)." <> just to check this - are you absolutely sure that you are the first ones to report this Swedish ice source in this area?

We are pretty certain, although we checked only English written literature. In fact Noormets and Flodén (2002b) state that area north of Gotska Sandon was an inter-stream area separating ice lobe located in the Landsort Deep from that in the Faro Deep. They however do not mention that each had a separate source. A Swedish ice source can be implicitly deducted from their study.

L_457-458: Bedforms within the Landsort Deep are not presented on map in Fig 4! The Deep is an interesting and outstanding submarine geological feature, so it would be good to discuss its origin (+ references) - could there be any alternative explanation for the features interpreted here as glacial lineations?

Our opinion is that this interpretation has a strong basis. Considering the quality of DEM (Fig 5G), context (continuation of glacial lineations further south and nearby inter-stream supported by Noormets and Flodén (2002b) ), sediments (partly coinciding with glacial clays) and geology (pre-quaternary sandstones contrary to surrounding hard bedrock supported by All et al. (2006)) these can be interpreted as glacial features. Greenwood et al. (2023 Boreas), independently, also interpreted these landforms as glacial lineations. The lineations are presented in Fig. 4.

Technical corrections

We thank the reviewer for going through the manuscript in detail and indicating technical corrections which we think should be included in text.

Technical corrections:

Technical points are corrected. Some more details are given below.

Figure 1: index map is of slightly poor resolution and the scale bar is missing from the main Figure 1B.

Resolution of Fig. 1 was slightly improved. However, the only higher resolution fig. is available as supplement S1. Scale is added.

Figure 1D: could the two troughs of Gotland Island be indicated on the NW-SE profile?

Done.

L_74: 1.1 <> should this rather be 2 Background?

Done.

Figure 4: use of colours for the lines is problematic even for slightly red-green blind reader (like me). In addition to colours also different line styles could be used to clarify visualization. Figure 4: "Light and dark grey indicate onshore and offshore areas respectively" <> I think this should be the other way around

With implementing dotted or dashed lines we will lost legibility, as some lines are very short. We changed colour of eskers to differentiate them from red moraines, for red-green blind readers. We think that the mapping for the main study area (Baltic Basin) is better visible on lighter background. For those not familiar with the Baltic Sea localisation, we provided Fig. 1 with details of the study area.

Figure 8: Again, in addition to colours, different line styles would improve readability! f.ex. uncertain observations could be marked with dashed lines? In southern Sweden there is one black line containing two different retreat directions? + Scale bar is missing!

With implementing dotted or dashed lines we will lost legibility, as some lines are very short. Two overlapping margins were separated.

Best regards,
Izabela Szuman-Kalita,
Jakub Kalita,
Christiaan Diemont,
Stephen Livingstone,
Chris Clark,
Martin Margold

**Dear Reviewer 2,**

We appreciate your comments on the preprint (in black). We believe that your contribution will lead to improvement in the quality of our manuscript. Please find our detailed response to your comments below (**in green**).

**General comments:**

This study is ambitious and the specific topic is highly relevant for the glacial dynamical development of the Fennoscandian ice sheet. The manuscript is of a suitable length and its structure is proper, making it easy to crasp. However, I agree with the comments of the anonymous referee and interactive comments by Greenwood et al. The mapping procedure quality issues (misinterpretations), lack of use of geological maps and the need for re-evaluation of some interpretations cause a need for major revisions.

The improved  version of our manuscript has some changes in landform mapping, slight changes in margin and flowset positions, according to geological maps we consulted and a newer version of EMODnet 2022 implementation  in mapping process. However our approach was focused on geomorphological expression of landforms not on their surface geology e.g., if the ridge met the requirements of moraine ridge (fan shaped, perpendicular to the ice flow direction and, in ideal situation, associated to other landforms) even if it was covered by e.g., marine sand it was classified as a moraine. We were not aiming for 'definitive geological survey standard' mapping, but rather to gather enough

information to act as a basis to build information about ice flow and ice margins. Indeed, we have now seen the recent publication of Greenwood et al. (2023, Boreas) - although these papers proceeded entirely independent of each other, it is apparent that there is a very large degree of similarity in the distribution, pattern and classification of landforms. Thus, we believe the landform misinterpretations were more technical problems that do not substantially affect the main conclusions of our paper, and which we retain.

We also included more alternative interpretations and refereed to the earlier studies on glacial landforms along Baltic Basin.

**Specific comments:**
(in order to minimize duplication, I will mainly comment only those points not mentioned by Greenwood et al. or Anonymous referee).

L25 (Abstract): "…broad changes in ice flow geometry, ranging from SE-NW to N-S and then to NWSE."
<> Ice flow direction replacing ice flow geometry? And maybe with description of the flow areas, for example: SE-NW in the western (SW of Malmö) area.

Finally, we decided that indication of each area is too broad as for the abstract (there are over 70 flowsets) and so describing all the areas is out of the abstract scope.

L139_L146: terminology to be clarified: The unconsolidated sediments versus sedimentary rocks. Also it is unclear what is meant with "soft sediments"

Agree, changed to unconsolidated when writing about sand and till, and sedimentary rocks when relevant.

L150: Elevation data => Bathymetric data?

Changed.

L205: Soft sediment => unconsolidated sediment?

Changed to '…soft sedimentary rocks and imprinted on thin drift..'

**Technical corrections:**

L77: "…from the south…"=> from towards the south
L134 and L138: "…north-western Baltic…" and "…SE Baltic…" => north-western Baltic main basin and SE Baltic main basin?

Changed

Figure 3B and 3D: conglometare => conglomerate (Is there a better word, for example bathymetric data type margin etc.?)

Changed to DEM stitching boundary

L429: could passed => could have passed

Changed.

Best regards,

Izabela Szuman-Kalita,
Jakub Kalita
Christiaan Diemont
Stephen Livingstone
Chris Clark
Martin Margold

---

## Referee Report (RR1)

**Reconstructing dynamics of the Baltic Ice Stream Complex during deglaciation of the last Scandinavian Ice Sheet**

Izabela Szuman1, Jakub Z. Kalita1, Christiaan R. Diemont2, Stephen J. Livingstone2, Chris D. Clark2, Martin Margold3

*1 Department of Geomorphology, Adam Mickiewicz University, Poznań, 5 62-712, Poland*
*2 Department of Geography, Sheffield University, Sheffield, S10 2NT, United Kingdom of Great Britain - England, Scotland,Wales*
*3 Department of Physical Geography and Geoecology, Charles University, Prague, 128 43, Czech Republic*

**General comments:**

The manuscript is now better written, easier to follow and more concise. The title is also better now. I still think that it is a bit difficult to filter the role of uncertain observations on maps but on the other hand I consider this point to be minor, not really affecting the outcomes or conclusions made. The main concerns I have raised earlier have been adequately noticed and answered. The fact that there has been simultaneous mapping within a part of the study area by Greenwood et al. 2024 does not seem to create marked contradictions but rather they work well together to improve our understanding of glacial dynamics and their role in future modelling approaches. Some minor specific comments and technical corrections are presented below.

**Specific comments:**

L34-35: Using the word amplitude in this context (Fig 1D) sounds odd to me (meaning relative height)

L77: Zeise, 1889; after Stephan, 2001) <> wrong reference style?  - should be Zeise, 1989 after Stephan, 2001)

Table 1: Referencess > References

L95+L108: reference to Greenwood et al. should be 2024 (ok in the reference list)

L138 v L139: terms glaciofluvial and glacifluvial are both used (also later in the manuscript), please prefer glaciofluvial.

Fig. 5B: I still don´t recognize these as ribbed moraines (see also Greenwood et al 2024) but this is a minor interpretative point in the wider context of the study)

Fig. 4: "Light and dark grey indicate onshore and offshore areas respectively" <> should be so that light grey are offshore (the deepest water)?

L275 (Fig. 5G): "This overlapping arrangement, seen here, is not common across the study area." <>  good point here – I think this Landsort Deep is a fascinating and enigmatic feature and the overwhelming presence of interpreted glacial lineations (also in Greenwood et al 2024) raises questions about their origin.

L317: > can be inferred

L345: Ice-marginal meltwater channels in the study are … (remove: in the study)

L404 Discussion: reference to work by Greenwood et al 2024 fits nicely here (but note the wrong year in the text reference)

L525-527: We confirm this … but also identify a Swedish ice source … <> I think the authors should refer here also to the findings by Greenwood et al. 2024

L566-568: Same here – reference to Greenwood et al 2024 needed or write differently to avoid using the reference in conclusions

---

## Author Response (AR2)

Dear Reviewer,

Thank you very much for careful going trough the manuscript. We included all your comments in the revised version. Also we provided better examples of ribbed moraines in Fig. 5I. Once again thank you very much.

Best regards,

Izabela Szuman with co-authors

---

## Author Response (AR3)

Dear Editor,

Thank you very much for careful going trough the manuscript. We included all your comments in the revised version.

Best regards,

Izabela Szuman with co-authors